

# Measurement report: Atmospheric new particle formation in a coastal agricultural site explained with binPMF analysis of nitrate CI-APi-TOF spectra

Miska Olin[1], Magdalena Okuljar[2], Matti P. Rissanen[1], Joni Kalliokoski[1], Jiali Shen[2], Lubna Dada[2,3], Markus Lampimäki[2], Yusheng Wu[2], Annalea Lohila[2,4], Jonathan Duplissy[2,5], Mikko Sipilä[2], Tuukka Petäjä[2], Markku Kulmala[2,6,7], and Miikka Dal Maso[1]

[1]Aerosol Physics Laboratory, Faculty of Engineering and Natural Sciences, Tampere University, Tampere, Finland
[2]Institute for Atmospheric and Earth System Research (INAR)/Physics, Faculty of Science, University of Helsinki, Helsinki, Finland
[3]Laboratory of Atmospheric Chemistry, Paul Scherrer Institute, Villigen, Switzerland
[4]Climate System Research, Finnish Meteorological Institute, Helsinki, Finland
[5]Helsinki Institute of Physics (HIP)/Physics, Faculty of Science, University of Helsinki, Helsinki, Finland
[6]Aerosol and Haze Laboratory, Beijing Advanced Innovation Center for Soft Matter Sciences and Engineering, Beijing University of Chemical Technology (BUCT), Beijing, China
[7]Joint International Research Laboratory of Atmospheric and Earth System Sciences, School of Atmospheric Sciences, Nanjing University, Nanjing, China

**Correspondence:** Miska Olin (miska.olin@tuni.fi)

**Abstract.** The occurrence of new particle formation (NPF) events detected in a coastal agricultural site, at Qvidja, in South-western Finland, was investigated using the data measured with a nitrate ion-based chemical-ionization mass spectrometer (CI-APi-TOF). The binned positive matrix factorization method (binPMF) was applied to the measured spectra. It resulted in eight factors describing the time series of ambient gas and cluster composition at Qvidja during spring 2019. The most inter-

esting factors related to the observed NPF events were the two factors with the highest mass-to-charge ratios, numbered 7 and 8, both having profiles with patterns of highly oxygenated organic molecules. It was observed that the factor 7 had elevated intensities during the NPF events. A variable with an even better connection to the observed NPF events is $f_{F7}$, which denotes the fraction of the total spectra within the studied mass-to-charge ratio range between 169 and 450 Th being in a form of the factor 7. Values of $f_{F7}$ higher than 0.5 were typically observed during the NPF events, of which durations also correlated with

the duration of $f_{F7}$ exceeding the critical value of 0.5. It was also observed that the factor 8 acts as a precursor of the factor 7 with high solar irradiance levels.

## 1 Introduction

Atmospheric new particle formation (NPF) and primary emissions are both important sources of the total particle number concentration and cloud condensation nuclei in the global troposphere and in the continental boundary layer (Merikanto et al.,

2009; Fountoukis et al., 2012; Posner and Pandis, 2015; Dunne et al., 2016; Kulmala et al., 2016; Gordon et al., 2017; Kerminen





et al., 2018; Olin et al., 2022). NPF occurring in different environments has a very diverse behavior and is also not well quantified (Kerminen et al., 2018).

Atmospheric particle measurements and related analyses of NPF events, i.e., events of the formation of new small particles and their subsequent growth, have been performed in a wide range of environments. NPF events seem to be occurring almost 20 everywhere in the world at all seasons, but their occurrence varies a lot between the sites and seasons (Kerminen et al., 2018). Probably the most intensively studied environments in terms of NPF are boreal forests and urban areas. In this study, we present the gas, particle, and cluster data measured in a coastal agricultural site and search for parameters favoring or disfavoring NPF events. NPF is often occurring as regional NPF events over large spatial areas of even hundreds of kilometers. On the other hand, it can be more local (Junninen et al., 2022).

Several parameters influencing the occurrence of NPF events have been proposed in the literature, but a consistent and universal parameterization is still lacking since the optimal parameters predicting NPF events in different locations vary significantly (Kerminen et al., 2018). Variables for predicting NPF most frequently reported are solar radiation (Birmili and Wiedensohler, 2000; Birmili et al., 2003; Guo et al., 2012; Jun et al., 2014; Pierce et al., 2014; Qi et al., 2015), pre-existing particle loadings (Birmili et al., 2003; Dal Maso et al., 2007; Pikridas et al., 2012; Salma et al., 2016; Dada et al., 2017), relative humidity (RH) 30 (Birmili et al., 2003; Wu et al., 2007; Guo et al., 2012; Jun et al., 2014), temperature ($T$) (Paasonen et al., 2013; Dunne et al., 2016; Dada et al., 2017), and the concentration of sulfuric acid ($H_2SO_4$) (Birmili et al., 2003; Kulmala et al., 2006; Wang et al., 2011; Qi et al., 2015; Yao et al., 2018) or its main precursor, sulfur dioxide ($SO_2$) (Birmili and Wiedensohler, 2000; Woo et al., 2001; Guo et al., 2012). Of those, high solar irradiances and high $H_2SO_4$ concentrations ($[H_2SO_4]$) typically favor the occurrence of NPF events. Instead, high pre-existing particle loadings, such as condensation sink (CS) consuming possibly 35 nucleating precursors, and high RH are typically disfavoring. Instead, $T$ and $SO_2$ concentration ($[SO_2]$) are inconclusive, as their roles in NPF events are ambiguous between different data sets. Also organic compounds with low enough volatilities or highly oxygenated organic molecules (HOMs), have recently been connected to atmospheric NPF (Dada et al., 2017; Bianchi et al., 2019). Specific for coastal environments are the roles of iodine compounds (O'Dowd et al., 2002), such as iodic acid ($HIO_3$) (Sipilä et al., 2016; Baccarini et al., 2020; He et al., 2021), and methanesulfonic acid ($CH_4SO_3$) (Beck et al., 2021) in 40 NPF. Ammonia ($NH_3$), as a base, is known to enhance $H_2SO_4$-driven nucleation through stabilizing critical clusters (Kulmala et al., 2014), which can be important in agricultural environments since it is a substantial component in fertilizers. Studies on NPF from agricultural land emissions have usually been related to $NH_3$, but also, e.g., skatole is suggested (Ciuraru et al., 2021), which is an organic compound in sewage sludge used in fertilizers.

Additionally, general criteria for the occurrence of NPF events have been searched for, but still, no suitable estimator covering 45 a variety of different environments has been found. Another difficulty with these criteria is that they usually require specific information on multiple variables, such as particle size distributions or growth rates, or precursor concentrations. In this study, we propose another parameter (named here $f_{F7}$) for predicting NPF events, which can be used in examining the occurrence of NPF events in coastal or agricultural areas or even in searching for the general criteria. Surprisingly, $f_{F7}$ does not include any particle variables but is solely based on mass spectrometric data of airborne molecules analyzed using the binned positive 50 matrix factorization method (binPMF) instead. More specifically, $f_{F7}$ denotes how much of the observed mass spectra is in a



specific form of the mixture of organic compounds. Additionally, the suggested explanation or routing of the observed NPF events gives insights into particle formation mechanisms occurring in the studied environment.

## 2 Measurement methods

### 2.1 Measurement site and time range

Measurements were performed at a pilot agricultural farm for regenerative farming, at Qvidja, located in a coastal environment in Turku Archipelago, in Southwestern Finland. The measurement site is located in the middle of fields and has the shortest distance to the sea of 500 m. The fields have clayey soil and they consist of several grass and clover species. Since 2017, sustainable and environmentally friendly field management practices have been conducted at the farm. More detailed information on the location, species, and management practices of Qvidja can be found in Heimsch et al. (2021).

The data used in this study are a part of long-term measurements provided by the instruments installed in a container in the middle of the fields. Additionally, a laboratory van, ATMo-Lab, was parked next to the container for the time range between 2 Apr 2019 and 26 Jun 2019. As the key instrument of this study, a mass spectrometer, was located in the van, only the data from this time range from the longer time series are utilized in this study.

### 2.2 Measurement instruments

The measurement instruments in the container include several gas analyzers, devices measuring environmental parameters, and aerosol samplers. Gas analyzers utilized in this study measure the concentrations of $SO_2$, ozone ($O_3$), carbon monoxide (CO), nitric oxide (NO), nitrogen oxides ($NO_x$), and $NH_3$. Utilized environmental parameters are temperature ($T$), relative humidity (RH), and net radiation (NetRad). The used aerosol samplers were two Particle Size Magnifiers (PSM,A and PSM,B), a Neutral Cluster and Air Ion Spectrometer (NAIS), and a Differential Mobility Particle Sizer (DMPS). PSM,A and PSM,B were used

to determine the particle size distributions in the size ranges of 1.15–2.8 nm and 1.3–2.8 nm, respectively. NAIS was used here to detect particles in the size range of 2.7–6.5 nm using its negative-polarity charger. DMPS was used to determine the particle size distribution in the size range of 6–823 nm.

A nitrate-ion-based ($NO_3^-$-based) chemical-ionization atmospheric-pressure-interface time-of-flight (CI-APi-TOF) mass spectrometer (Aerodyne Research Inc.; USA and Tofwerk AG, Switzerland; Jokinen et al. 2012) was installed in the labo-

ratory van. It consists of a chemical-ionization inlet (Eisele and Tanner, 1993) and an APi-TOF mass spectrometer (Junninen et al., 2010). The air sample was drawn with the flow rate of 8.3 slpm through a side hatch of the van with a 16 mm pipe of 60 cm length.

## 3 CI-APi-TOF data processing

The data were recorded in 4 s time resolution but were first averaged to 600 s resolution ($t_a$) with a TOF-mass spectrometer

data processing code for Matlab, tofTools, resulting in 9970 valid time bins. The averaged TOF-spectra were then calibrated





with the tofTools code against known mass-to-charge ratios ($m/z$) of always existing ions with $NO_3^-$-based CI-inlet, $NO_3^-$, $NO_3^- \cdot HNO_3$, and $NO_3^- \cdot (HNO_3)_2$ or $NO_3^- \cdot$nitrophenol ($C_6H_4OHNO_2$).

### 3.1 Potential new particle-forming acids

Compounds previously most associated with NPF observed in a coastal environment measured using the CI-APi-TOF are the
acids $H_2SO_4$, $HIO_3$, and $CH_4SO_3$.

[$H_2SO_4$] is calculated as by Olin et al. (2020) with the equation

$$[H_2SO_4] = \qquad\qquad\qquad\qquad\qquad\qquad\qquad\qquad\qquad\qquad\qquad (1)$$
$$\frac{C}{P} \cdot \frac{\{HSO_4^-\} + \{HSO_4^- \cdot HNO_3\} + \{HSO_4^- \cdot H_2SO_4\}}{\{NO_3^-\} + \{NO_3^- \cdot HNO_3\} + \{NO_3^- \cdot (HNO_3)_2\}}$$

where $C$ is the calibration coefficient for $H_2SO_4$ ($= 2.45 \times 10^9 \, cm^{-3}$) determined with known $H_2SO_4$ concentrations (details
of the method are well described in Kürten et al. (2012)) at the measurement site, $P$ is the penetration efficiency of $H_2SO_4$
in the sampling pipe ($= 0.58$), and the curly brackets denote the areas of the peaks at the corresponding mass-to-charge ratios
in the high-resolution mass spectra obtained from the tofTools code. Although generally untypical, there was another peak
overlapping with the bisulfate ion, $HSO_4^-$, peak (96.96 Th) at 96.97 Th in these data. The overlapping peak corresponds to
$H_2CO_3 \cdot Cl^-$ (hydrochloric acid, HCl, and carbonic acid, $H_2CO_3$, were also detected separately), and it covered 10–90 %
of the area of the total peak at 97 Th. Therefore, high-resolution fitting was a necessity in determining [$H_2SO_4$] in this case,
whereas fitting with the unit mass resolution (UMR) is sufficient in many cases. The detection of $H_2CO_3$ with $NO_3^-$ ionization
is untypical and it usually implies insufficient $HNO_3$ in the ionizer. This potentially leads to detecting less bound product$\cdot NO_3^-$
adducts, i.e., decreasing the selectivity (Hyttinen et al., 2015), yet it is not an easy task to estimate how much. Nothing such
was observed but it should be kept in mind that the data from the last 3 weeks (during which the $H_2CO_3$ signal was at its
highest) has to be interpreted with caution.

The concentrations of $HIO_3$ ([$HIO_3$]) and $CH_4SO_3$ ([$CH_4SO_3$]) were calculated as for $H_2SO_4$, with the exception of the
peaks $\{IO_3^-\} + \{IO_3^- \cdot HNO_3\}$ and $\{CH_3SO_3^-\} + \{CH_3SO_3^- \cdot HNO_3\}$, respectively, as a numerator in Eq. (1). Due to the
lack of calibration methods for compounds other than $H_2SO_4$, we used a common approximation in which the values of $C$
and $P$ determined for $H_2SO_4$ were used for these compounds too. That is because these species ($HIO_3$ and $CH_4SO_3$) have
collision-limited charging efficiency when reacting with $NO_3^-$ ions (Simon et al., 2020; Beck et al., 2021; Wang et al., 2021).

### 3.2 binPMF analysis

The mass spectra between 169 and 450 Th obtained from the tofTools code were analyzed using the binPMF analysis method
described by Zhang et al. (2019). Ions smaller than 169 Th were omitted because there are many organic compounds that are
unlikely the key compounds in NPF and have relatively high signals possibly causing issues in the binPMF analysis. One of
the highest peaks in the spectra, malonic acid-$NO_3^-$-cluster (166 Th), with its isotopes, were the largest of the omitted ions.
Ions larger than 450 Th were also omitted due to their reduced transmission efficiency inside the API-TOF device with the





used voltage settings. Additionally, ions between 188 and 190 Th were omitted because the nitrate trimer—one of the reagent ions—with its isotopes falls in that range.

As by Zhang et al. (2019), the spectra were binned to bins with a width of 0.02 Th between N−0.2 and N+0.3 Th where N is an integer mass, resulting in 25 bins per integer, i.e., in 6975 bins in total in this case. The basis of PMF, in general, is to express the measured data matrix (X) with a time series matrix of factor intensities (TS), a factor mass spectra matrix (MS), and a residual matrix (R). The residuals are tried to be minimized so that the factor construction would describe the measured spectra most realistically. The number of factors, $p$, is a free parameter and its optimal value can be estimated by minimizing the residuals. In this case of binPMF, this is mathematically expressed with the equation

$$X_{9970 \times 6975} = TS_{9970 \times p} \times MS_{p \times 6975} + R_{9970 \times 6975}. \tag{2}$$

The data matrix X was normalized before running the binPMF code with the reagent ion signals as in Eq. (1).

The binPMF code tries to find the matrices TS and MS producing the lowest possible value for the sum of the scaled residuals,

$$Q = \sum_{i=1}^{9970} \sum_{j=1}^{6975} (R_{ij}/S_{ij})^2, \tag{3}$$

where S is the uncertainty matrix. S was estimated from the ambient data via the method suggested by Yan et al. (2016), i.e., through approximating instrument noise as the difference between the measured signal and its moving median over 5 data points. Assuming that the counting statistics follow the Poisson distribution, the function

$$S_{ij} = \sigma_{ij} + \sigma_{\text{noise}} = a\sqrt{\frac{\max\left(I, \frac{1}{t_a}\right)}{t_a}} + \sigma_{\text{noise}} \tag{4}$$

$$= (1.35 \pm 0.22)\sqrt{\frac{\max\left(I, \frac{1}{600\,\text{s}}\right)}{600\,\text{s}}} + (0.001 \pm 0.003)\,\text{cps}$$

where $I$ is the signal intensity in counts per second (cps), was obtained. The correcting factor $a$ incorporates any unaccounted contributions to the uncertainty (Allan et al., 2003). The value of $a$, $1.35 \pm 0.22$, is on a similar level to the values in the study by Yan et al. (2016), $1.28 \pm 0.09$ or $1.1 \pm 0.3$. Similar to the matrix X, the matrix S was also normalized with the reagent ion signals before running the binPMF code.

The binPMF code used in this study (customized Matlab's nnmf function) includes similar down-weighting schemes for 135 signals with low signal-to-noise ratios and for outliers with $|R_{ij}/S_{ij}| > 4$ as the code used by Zhang et al. (2019). A theoretically expected $Q$ value ($Q_{\text{exp}}$) can be calculated as the number of non-down-weighted elements in X subtracted by the number of elements in TS and MS in total. The code was run with increasing $p$ until the ratio $Q/Q_{\text{exp}}$ did not substantially decrease anymore. Because a PMF analysis depends on the initial guesses of TS and MS (seeds), the code was run with 50 random seeds per every $p$ to find the optimal $p$ and finally with 300 random seeds with the optimal $p$, of which the seed providing the lowest 140 $Q$ value was selected for further analysis.

Because the factor intensity matrix TS is a dimensionless variable, it is further converted to a practical variable using the calibration coefficient and penetration efficiency of $H_2SO_4$ also for the factor intensities. Hence, the factor intensities denote





hereafter the total concentrations of all compounds within a factor. However, it should be noted that a total concentration of a factor would be true only if all the compounds would have responses between concentrations and the normalized signals and

the penetration efficiencies similar to $H_2SO_4$.

## 4   Results and discussion

From all 84 measurements days, 25 days show clear features of NPF events, during which a new particle mode appears by growing from the very small particle sizes near 1 nm toward a larger-sized background mode and finally merges with it. Instead, 37 days have hardly any NPF event features and 19 days have unclear features and cannot thus be classified as event

or non-event days. The remaining 3 days remain unsure due to gaps in the measurement data. These classifications are listed in Table S1.

### 4.1   Time series of measured variables

Figure 1 presents time series of particle size distribution contours with variables most likely promoting NPF for three example time ranges. The time range before the fertilization (Fig. 1a) represents a great example of a quite constant background particle

mode around 100 nm during the first three days followed by a clear and strong NPF event (Apr 28th). The subsequent three days also have NPF event features but they become weaker every day and the last one (May 1st) has just hints of a very short NPF event. The reason for the inexistent NPF during the first three days is probably the suppression due to a high CS level ($0.01 - 0.02\,1/s$). The reason for the strong NPF event on Apr 28th and the subsequent weakening events cannot be clearly explained with any of the measured variables. Nevertheless, the most promising particle-forming compound measured

is $H_2SO_4$ because its concentration is elevated approximately at the same time as the events. However, it does not show a decreasing trend during the weakening trend of the NPF events and CS also remains on a lower level ($\sim 0.005\,1/s$). $[HIO_3]$ is also elevated during the daytime but its levels do not explain these NPF events because it is its highest on the last two days, having no clear NPF events. $[NH_3]$ and $[CH_4SO_3]$ seem to not clearly explain any changes in particle size distribution data.

    The time range during the fertilization (May 8th) and right after it (Fig. 1b) includes one of the strongest NPF events (May

14th) during this measurement. Noteworthy is that the fertilization increases daytime $[NH_3]$ remarkably (100-fold compared to the time right before the fertilization and 10-fold compared to the average of the preceding days). However, the occurrence of NPF events does not increase after the fertilization even though $[NH_3]$ remains high, implying that, on one hand, $NH_3$ is not a key compound or is not a limiting factor in NPF events during these measurements. On the other hand, $[NH_3]$ seems to be higher during the days showing NPF features, with the exception of the last days, during which CS begins to approach and

exceed the level of $0.01\,1/s$ showing high $[NH_3]$ but no NPF. $[H_2SO_4]$ seems promising here too but still no clear criterion for the occurrence of an NPF event at this site cannot be constructed from any of the measured variables.

    The time range during the harvest (Fig. 1c) shows another type of changes in particle size distribution data. There are short spikes of increased particle concentrations within the size range of 20–200 nm and thus in CS too, occurring several times a day. They originate probably from direct emission sources, e.g., from tractors harvesting the fields, rather than via NPF because

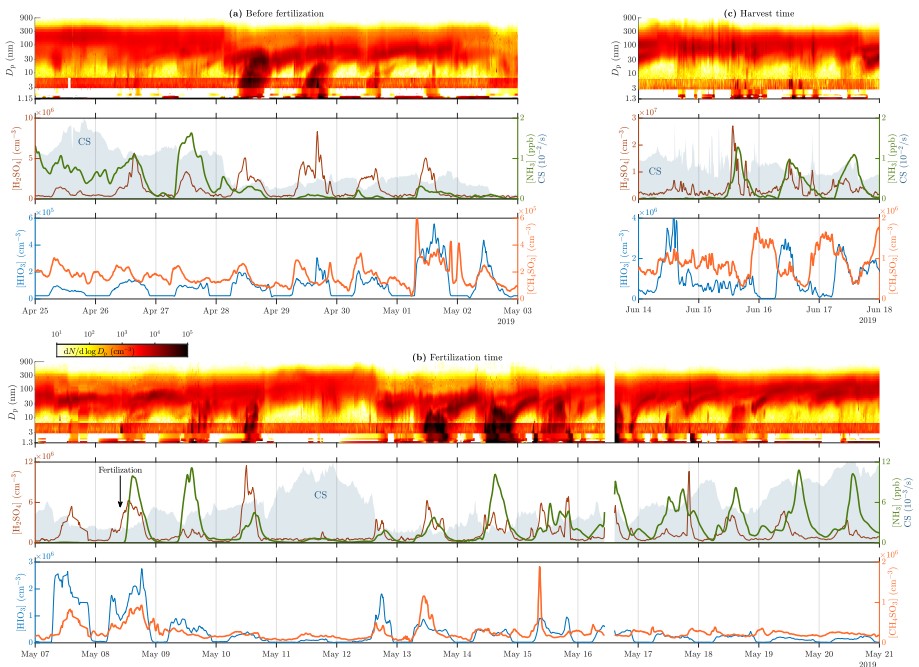

**Figure 1.** Time series of particle size distribution contours (top panels), $[H_2SO_4]$, $[NH_3]$, CS (shaded areas in the middle panels), $[HIO_3]$, and $[CH_4SO_3]$ (bottom panels) for different example time ranges, **(a)** before the fertilization, **(b)** during and right after the fertilization, and **(c)** during the harvest. Particle data below 3 nm are from a PSM, over 6 nm are from the DMPS, and the remaining part between them is from the NAIS. Note that the PSM data shown in **(a)** are from the PSM,A whereas the data shown in **(b)** and **(c)** are from the PSM,B due to different coverage of the devices. Also, note different scales of the y-axes between the subfigures.

the abundance of the smallest particles is simultaneously not elevated. It can be observed that also the $[H_2SO_4]$ time series contains spikes at the coincident time moments, highlighting the possibility of tractors as the source of 20–200 nm particles.

## 4.2 binPMF results

Previously presented analysis on the measured variables did not provide any clear formula for the occurrence of the NPF events in this measurement site. Next, we concentrate on the results from the binPMF analysis and how they can be related to the occurrence of the events.

### 4.2.1 Determining the optimal number of factors

Figure 2 presents the lowest obtained ratios $Q/Q_{\exp}$ with different random seeds and with different numbers of factors ($p$). It can be seen that $Q/Q_{\exp}$ decreases with increasing $p$, as expected. The decrease rate is at its greatest when $p$ increases from 2 to 6. This denotes that there is a significant improvement in every step in which the measured spectra are described with an additional factor beginning from the set of two factors only. It also denotes that there could be six main sources resulting





in the measured spectra. However, $Q/Q_{\exp}$ still decreases slightly from $p = 6$ to $p = 9$, denoting that taking more than six sources into account provides a slight improvement in describing the spectra with more factors. The decreasing rate of $Q/Q_{\exp}$ diminishes almost totally after $p = 9$ and $Q/Q_{\exp}$ sets to 4. Theoretically, $Q/Q_{\exp}$ should set to 1 if the uncertainty matrix S has been constructed ideally. Thus, $Q/Q_{\exp}$ of 4 being over 1 denotes that underestimations exist in estimating the uncertainties

of the measured spectra. According to the decrease rates, the optimal $p$ would be in the range of 6–9. The lowest sensible $p$ is preferred because it simplifies further analysis. Because organic compounds are in a key role in this study, the value $p = 8$ was selected for further analysis because it is the lowest value producing the MS matrix which includes at least 4 different factor profiles having organic patterns separated with 14–16 Th (3 factor profiles having organic patterns with $p = 6$ or $p = 7$). These separations correspond to the increments of mass due to additional oxygen molecules or due to different carbon chain lengths.

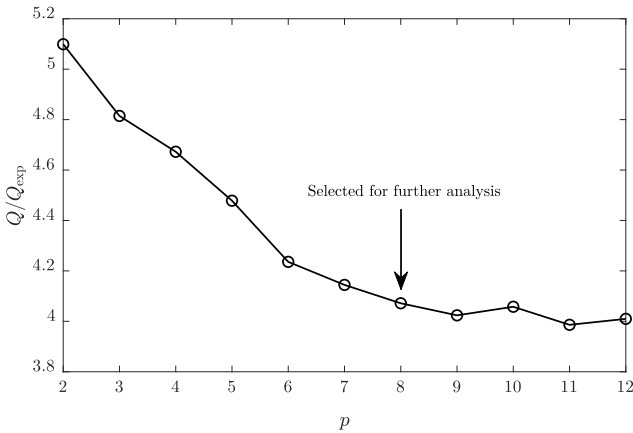

**Figure 2.** The lowest obtained $Q/Q_{\exp}$ as a function of the number of factors ($p$).

### 4.2.2 Selected set of factors

Figure 3 presents an overview of the factor profiles (arranged according to their average masses) and Fig. 4 their diurnal variations obtained with $p = 8$. It can be observed with a rough examination that the factors 4, 5, 7, and 8 have organic patterns. In addition to those, factors 2 and 6 have sharp peaks repeating with 18 Th, which correspond to the nitrate monomer and dimer clustered with different numbers of water molecules. Conversely, the remaining factors, 1 and 3, have peaks located with no

clear patterns. The diurnal variations show that the factors 1, 3, 4, and 7 are typically encountered in the daytime, while the rest are at their highest during mornings.

The key strength of a binPMF analysis is its applicability in examining the factor profiles with high mass resolution. A more thorough examination of the factor profiles and their naming is performed next. Several peaks were identified from the profiles using the tofTools code and the highest ones in every profile are presented in Table 1. A more extensive peak list can be found

in Table S2. It is worthwhile to note that the $m/z$ axis was calibrated using the known compounds within the range of 62–201 Th, causing that the peaks at the higher end of the spectra are systematically slightly shifted toward larger values. Additionally,

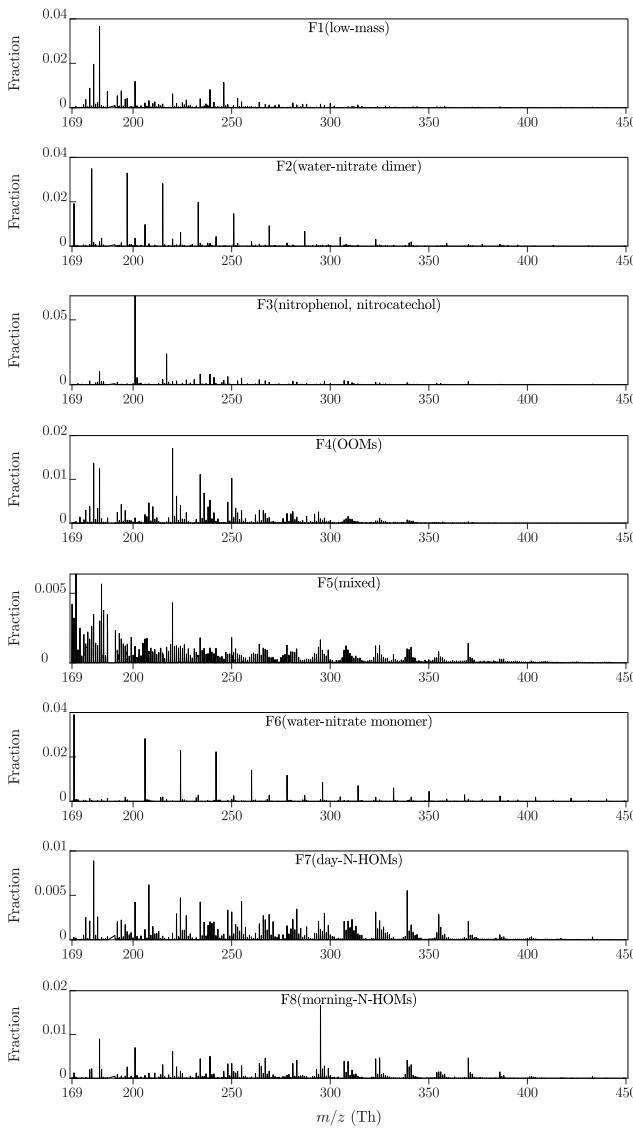

**Figure 3.** Factor profiles normalized with the total factor signal, i.e., the sum of the signal from every $m/z$ bin of a factor is 1. Zoomed views of these spectra can be found in Figs. S1–S8 and their raw spectra are also available at Olin (2022).





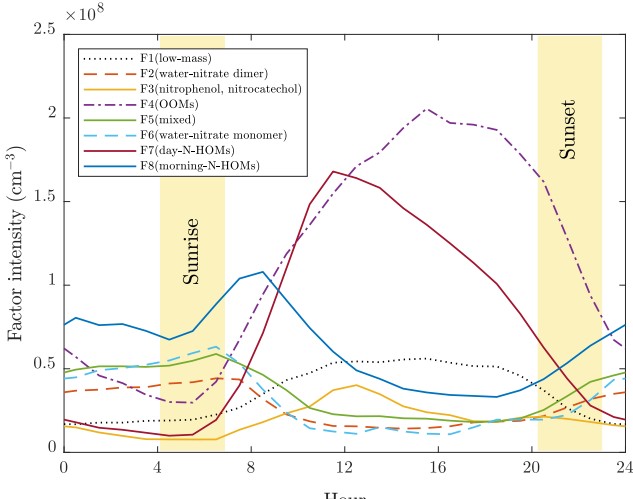

**Figure 4.** Mean diurnal variations of the factor intensities. The yellow areas denote the variation in sunrise and sunset times during the experiments.

due to the limited mass resolution of a binPMF analysis, the spectra in the factor profiles can be further dispositioned slightly. Nevertheless, identifying possible chemical formulae from the observed peaks was still done with relatively high confidence for several peaks. Note that determining correct isomers of identified chemical formulae is generally impossible with the
TOF-mass spectrometer.

### 4.2.3 Characteristics of the factors 2 and 6

The factors 2 and 6 are the factor profiles which include the clearest and most confidently identified peaks and their repeating pattern. The factor profiles consist almost solely of the clusters of nitrate dimer (the factor 2) or nitrate monomer (the factor 6) and 3–16 (the factor 2) or 6–21 (the factor 6) water molecules. They also have similar morning-type diurnal patterns but
are still separated into different factors because the factor 2 has a higher intensity in the beginning part of the measurement period and the factor 6 vice versa. These water clusters are instrumental impurities caused by humidity inside the CI-inlet, where water molecules cluster with newly formed nitrate ions. In this article, these factors are named *F2(water-nitrate dimer)* and *F6(water-nitrate monomer)*.

Because the water cluster-peaks in F2(water-nitrate dimer) and F6(water-nitrate monomer) are detected with very high
confidence up to 400 Th, their errors can be used as another type of mass calibration, performed after the binPMF analysis. The water cluster-peaks have errors between -10 ppm and 0 ppm when $m/z$ is below 280 Th and increase from 0 ppm to +50 ppm when $m/z$ increases from 280 Th to 400 Th. Hence, also the peaks in the other factors are expected to have similar error levels.





**Table 1.** The highest peaks in the binPMF factors. Observed $m/z$ ratios[a], chemical formulae corresponding best to them (in neutral forms, i.e., without the nitrate ion or with an added proton)[b], errors between the observed and exact $m/z$ ratios[c], possible compound names[d], and confidence levels of the identifications[e] are presented. A more extensive peak list can be found in Table S2.

| F1 (low-mass) | F2 (water-nitrate dimer) | F3 (nitrophenol, nitrocatechol) | F4 (OOMs) | F5 (mixed) | F6 (water-nitrate monomer) | F7 (day-N-HOMs) | F8 (morning-N-HOMs) |
|---|---|---|---|---|---|---|---|
| [a] 182.9960 Th | 179.0170 Th | 201.0171 Th | 220.0465 Th | 171.0606 Th | 170.0531 Th | 180.0156 Th | 295.0749 Th |
| [b] $C_7H_4O_6$ | $(H_2O)_3HNO_3$ | $C_6H_4OHNO_2$ | $C_7H_{10}O_4$ | unidentified | $(H_2O)_6$ | $C_4H_6O_4$ | $C_9H_{15}O_6N$ |
| [c] -14 ppm | -7.7 ppm | -9.3 ppm | -1.5 ppm | | -8.4 ppm | -4.0 ppm | +11 ppm |
| [d] chelidonic acid | water·nitrate dimer | nitrophenol | very high | | water·nitrate | succinic acid | moderate |
| [e] low | very high | very high | | | very high | very high | |
| 180.0171 Th | 197.0278 Th | 217.0109 Th | 180.0173 Th | 183.9966 Th | 206.0747 Th | 208.0159 Th | 182.9943 Th |
| $C_4H_6O_4$ | $(H_2O)_4HNO3$ | $C_6H_3(OH)_2NO_2$ | $C_4H_6O_4$ | unidentified | $(H_2O)_8$ | $C_5H_6O_5$ | $C_7H_4O_6$ |
| -12 ppm | -7.8 ppm | -3.3 ppm | -13 ppm | | -9.0 ppm | -29 ppm | -4.9 ppm |
| succinic acid | water·nitrate dimer | nitrocatechol | succinic acid | | water·nitrate | ketoglutaric acid | chelidonic acid |
| high | very high | high | high | | very high | low | moderate |
| 201.0153 Th | 215.0387 Th | 182.9971 Th | 182.9899 Th | 220.0135 Th | 224.0849 Th | 339.0585 Th | 201.0159 Th |
| $C_6H_4OHNO_2$ | $(H_2O)_5HNO_3$ | $C_7H_4O_6$ | $C_7H_4O_6$ | unidentified | $(H_2O)_9$ | $C_{10}H_{15}O_8N$ | $C_6H_4OHNO_2$ |
| -0.4 ppm | -8.9 ppm | -20 ppm | +19 ppm | | -6.5 ppm | +28 ppm | -3.4 ppm |
| nitrophenol | water·nitrate dimer | chelidonic acid | chelidonic acid | | water·nitrate | moderate | nitrophenol |
| very high | very high | moderate | low | | very high | | very high |
| 246.0000 Th | $(H_2O)_{6...16}HNO_3$ | 234.0615 Th | $C_{8...9}H_{12...14}$ | OOMs, HOMs, | $(H_2O)_{10...21}$ | $C_{7...10}H_{11...15}$ | $C_{7...10}H_{11...15}$ |
| unidentified | | $C_8H_{12}O_4$ | $O_{4...6}$ | and N-HOMs | | $O_{6...9}N$ | $O_{6...9}N$ |
| | | +1.7 ppm | | 240–390 Th | | | |
| | | terpenylic acid | | | | | |
| | | moderate | | | | | |

### 4.2.4 Characteristics of the factor 3

The factor 3 comprises mainly of a peak typically encountered with nitrate ionization, $NO_3^- \cdot C_6H_4OHNO_2$ (nitrophenol, $C_6H_4OHNO_2$, clustered with a nitrate ion), and a peak at 217.0109 Th. The latter one is possibly from $C_6H_3(OH)_2NO_2$, which corresponds to nitrocatechol (or its isomer). Both nitrophenol and nitrocatechol are connected at least to biomass burning (Iinuma et al., 2010). Additionally, other peaks located with no clear pattern, however, with much lower intensities exist in the factor 3. The ones of these with the highest intensities include peaks at 182.9971 Th and 234.0615 Th, which correspond best

with $C_7H_3O_6^-$ and with $NO_3^- \cdot C_8H_{12}O_4$, respectively. One possibility is that they are from chelidonic and terpenylic acid, both connected to vegetation. This factor is named here *F3(nitrophenol, nitrocatechol)*.





### 4.2.5 Characteristics of the factor 4

The next clearest of the factor profiles is the factor 4. It consists mainly of oxidized organic molecules (OOMs) within the $m/z$ range of 200–350 Th (including clustered nitrate ions, which are hereafter omitted from the shown chemical formulae for convenience) having the chemical formulae of a form of $C_xH_yO_z$, of which the peaks identified with high confidence have x of 7, 8, or 9, y of 10, 12, or 14, and z of 4, 5, or 6. The largest of these molecules can also be considered as HOMs, by the definition of a HOM having at least six oxygen atoms (Bianchi et al., 2019). Additionally, the factor 4 includes peaks corresponding likely with succinic acid and possibly with chelidonic acid (at 182.9899 Th). Succinic acid together with peaks with lower intensities and lower confidence levels also likely found, fumaric and malic acid, belong in the citric acid cycle, which is a metabolic pathway of aerobic organisms, including plants. However, the peak at 182.9899 Th can potentially be something other than chelidonic acid, at least a different compound than in other factors having a peak at ∼183 Th, because the peak in this factor falls on the other side of the exact mass of deprotonated chelidonic acid (182.9935 Th) compared to the other factors. A lower error than for chelidonic acid (+19 ppm) is achieved with $C_2H_3O_5N$ (-3 ppm), which could be peroxyacetyl nitrate but its detection with $NO_3^-$-based ionization has not been reported before. This factor is named here *F4(OOMs)*.

### 4.2.6 Characteristics of the factors 7 and 8

The factors 7 and 8 have repeating organic patterns (mainly N-HOMs which are HOMs with an extra nitrogen atom) in the $m/z$ range of 230–410 Th. These N-HOMs have the chemical formulae of a form of $C_xH_yO_zN$, of which the peaks identified with high confidence have x of 7, 8, 9, or 10, y of 11, 13, or 15, and z of 6, 7, 8, or 9. Though the transmission efficiency decreases with increasing $m/z$, a high peak at 339.0585 Th being likely $C_{10}H_{15}O_8N$ in the factor 7 and a very high peak at 295.0749 Th being likely $C_9H_{15}O_6N$ in the factor 8 are distinguished clearly from the profiles. The first indication of the connection between $C_{10}H_{15}O_8N$ and atmospheric nucleation was reported by Kulmala et al. (2013).

The profiles of the factors 7 and 8 differ also in a way that the peaks at the lower end of the $m/z$ range are of higher intensity in the factor 7 than in the factor 8. These peaks in the factor 7 include also compounds that can be related to the citric acid cycle, e.g., succinic, ketoglutaric, and malic acid. Instead, no citric acid cycle-related compounds exist in the factor 8 but nitrophenol and possible chelidonic acid do. According to their diurnal patterns, these factors are named here *F7(day-N-HOMs)* and *F8(morning-N-HOMs)*.

### 4.2.7 Characteristics of the factor 5

The factor 5 is elevated at nighttime and consists of several unidentified peaks in the lower end of the $m/z$ range and of OOMs, HOMs, and N-HOMs mixed with each other in the $m/z$ range of 240–390 Th. No clear pattern can be observed from this mixture.

One key feature in the factor 5 is also its clearly increasing intensity toward the summer, while the other factors do not show that kind of behavior. Although with low intensity and confidence, there are also peaks suggesting citric acid cycle-related compounds, e.g., fumaric, malic, and aconitic acid. Additionally, although not directly included in the factor 5, pyruvic acid (at





87.0088 Th in the spectra before binPMF), a key compound in the beginning of the citric acid cycle, seems to be encountered

simultaneously with the factor 5. This factor is named here *F5(mixed)*.

### 4.2.8  Characteristics of the factor 1

The last one of the factor profiles, the factor 1, is elevated at daytime and consists mostly of low-mass compounds, including nitrophenol, succinic acid, probably chelidonic and malic acid, and several other peaks without any clear pattern between them. This factor is named here *F1(low-mass)*.

## 4.3  Correlations between variables

Figure 5 presents Pearson's correlation coefficients ($R$) between all possible pairs of measured variables and binPMF factors in 10 min time resolution. Connecting any variable to NPF events can be estimated with correlations to particle variables, such as to the number concentration ($N$) measured by PSMs, NAIS, or DMPS, and to CS. It should be noted that this is only an estimation on the connections between variables and NPF events. Thus, it cannot be certainly proved that a variable is

actually forming new particles or growing them by examining the correlations. There is always a possibility that a variable is only observed simultaneously with NPF events due to the similarity of its source and the source of the precursor really causing the NPF events. In Fig. 5, the number concentrations from the different devices represent different particle size ranges; additionally, $N_{\mathrm{NAIS}}^{-}$ refers only to particles detected using the negative-polarity charger. It can be seen that clearly positive correlations for all particle sizes are observed with F3(nitrophenol, nitrocatechol), F7(day-N-HOMs), [SO$_2$], [NH$_3$], [H$_2$SO$_4$],

$T$, and NetRad, of which F3(nitrophenol, nitrocatechol) and [H$_2$SO$_4$] have the strongest ones. It is clear why $T$ and NetRad are positively correlated with particles because NPF events almost exclusively occur during the daytime. The connections of [NH$_3$] and [H$_2$SO$_4$] to NPF were observed already with Fig. 1b, which also explains the correlation of [SO$_2$] because H$_2$SO$_4$ is formed from SO$_2$. The strongest negative correlations are observed with RH, which is obvious due to the fact that NPF events generally occur mostly with low RH.

The correlations between the binPMF factors and the other variables give hints for the contents and sources of the factors. F1(low-mass) correlates relatively well with [O$_3$], suggesting that F1(low-mass) includes compounds related to ozonolysis. F2(water-nitrate dimer) and F6(water-nitrate monomer) correlate well with RH because they consist mainly of the water clusters of nitrate, which are more abundant in high RH. F3(nitrophenol, nitrocatechol) correlates well with [SO$_2$], [CO], [NO$_x$], and [H$_2$SO$_4$], which suggests that F3(nitrophenol, nitrocatechol) could be originated from combustion sources, such as from

tractors, as already discussed before. F4(OOMs) has the highest correlations with [O$_3$] and [NH$_3$] and also a very high correlation with $T$. A deeper examination between F4(OOMs) and $T$ shows that an even higher correlation is achieved with a vapor pressure-type function of $T$, $\exp\left(AT + B\right)$, which implies that the origin of F4(OOMs) could be related to the evaporation of some compounds, possibly requiring O$_3$ to be detected as HOMs with the CI-APi-TOF. F5(mixed) correlates well with the concentrations of inorganic compounds, such as with [NH$_3$], [H$_2$SO$_4$], [HIO$_3$], and [CH$_4$SO$_3$].

F7(day-N-HOMs) correlates well with [O$_3$], [NH$_3$], and [H$_2$SO$_4$] whereas F8(morning-N-HOMs) correlates well with [NO$_x$]. Their strongest correlations, however, are against $T$, RH, and NetRad, of which $T$ correlates positively with both



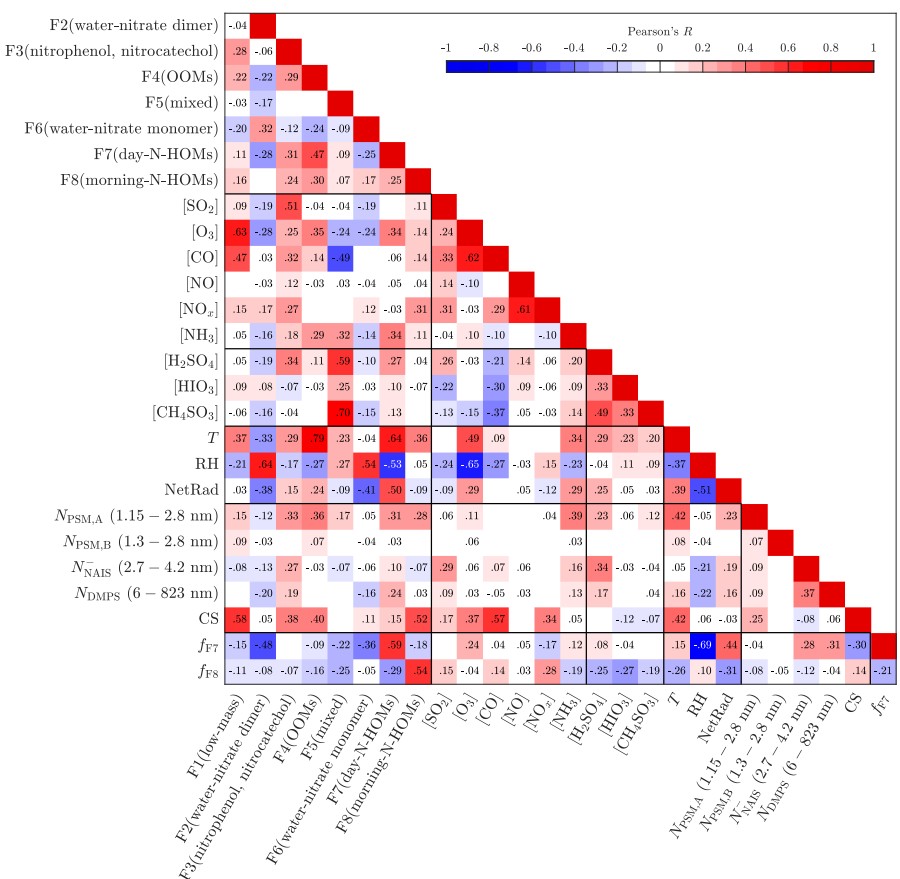

**Figure 5.** Pearson's correlation coefficient ($R$) matrix between measured variables and binPMF factors. Values of $R$ are shown as colors and numerically. They are shown only for statistically significant pairs (p-value below 0.05).

F7(day-N-HOMs) and F8(morning-N-HOMs), but RH and NetRad behave differently. F7(day-N-HOMs) has a strong negative correlation with RH and a strong positive correlation with NetRad, whereas F8(morning-N-HOMs) has weak correlations with RH and NetRad in the opposite directions. F7(day-N-HOMs) and F8(morning-N-HOMs) are further examined by defining variables $f_{F7}$ and $f_{F8}$, which refer to their fractions in the total spectrum. In other words, e.g., $f_{F7}$ denotes how much of the spectrum is in an F7-like form. A positive correlation of $f_{F7}$ with NetRad and a negative correlation of $f_{F8}$ with NetRad denote that the spectrum prefers the F7-like form over the F8-like form with higher NetRad levels. This can be interpreted so that N-HOMs are in the F8-like form in the mornings but solar radiation transforms them into the F7-like form for daytime.

Particle concentrations are expressed with only 4 size ranges (total concentrations from PSM,A, PSM,B, NAIS, and DMPS) and with CS in Fig. 5. The connection between NPF events and the measured variables can be further examined by expressing the correlation coefficients for all particle sizes (Fig. 6a). Notable is that the correlation coefficients for $[H_2SO_4]$, $[NH_3]$, and $[SO_2]$ are positive from the smallest particle sizes up to around 30 nm. It suggests that they are involved in forming new





particles and growing them to those sizes but not to larger sizes because the correlation coefficients approach zero near 30 nm.
It is known that high NetRad and low RH favor NPF events. This can be observed in Fig. 6a as positive and negative correlation
coefficients, respectively, from the smallest sizes up to ∼60 nm, which can be interpreted as a rough validation of this method
in examining connections between variables and NPF events. The most interesting particle size range in terms of a NPF event
in this case is around 10–40 nm because the background aerosol is usually in larger sizes and the newly formed particles in
smaller sizes. Particle concentrations within this size range increase mainly during NPF events only; the size distributions of
the newly formed particles sweep past this size range (see Fig. 1). In Fig. 6a, only the correlation coefficient for NetRad stays
clearly positive (and clearly negative for RH) within the size range of 10–40 nm.

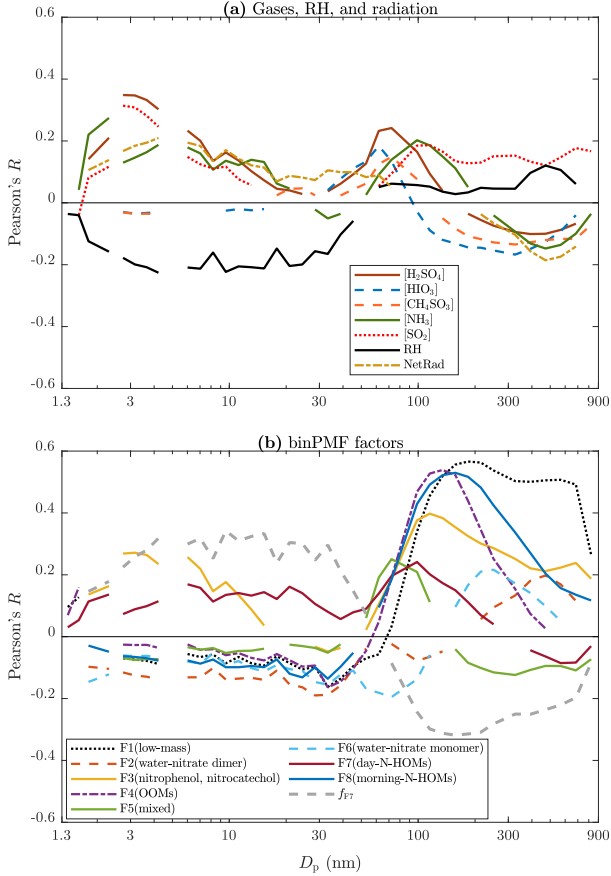

**Figure 6.** Pearson's correlation coefficient ($R$) between all particle size bins and **(a)** measured variables and **(b)** binPMF factors. Data below
3 nm are from a PSM, over 5 nm are from the DMPS, and the remaining part between them are from the NAIS. Lines are shown only for
statistically significant pairs (p-value below 0.05).

Figure 6b presents the correlation coefficients for the binPMF factors as a function of particle size. F3(nitrophenol, nitro-
catechol) seems to be somehow involved in forming new particles but not in growing them past the 10–40 nm size range. The





correlation coefficient for F7(day-N-HOMs), instead, stays on the positive side even up to 300 nm, while for all other factors they are on the negative side within the lower half of the whole particle size range. Within the size range of 10–40 nm, F7(day-N-HOMs) is the only one of the factors having positive correlation coefficients. Even stronger correlations are achieved when examining the fraction of F7(day-N-HOMs) in the total spectrum, $f_{F7}$. From these plots (Fig. 6), it can be hypothesized that [H$_2$SO$_4$], [NH$_3$], F3(nitrophenol, nitrocatechol) and F7(day-N-HOMs) participate in forming new particles but only F7(day-N-HOMs) is related to growing the particles large enough to provide a full NPF event. Figure S9 presents similar plots for [H$_2$SO$_4$], [NH$_3$], F3(nitrophenol, nitrocatechol), F7(day-N-HOMs), and $f_{F7}$ separately for days with and without NPF events. It can be seen that correlations are stronger during the days with NPF events for particle sizes up to ~40 nm.

## 4.4 Time series with binPMF factors

Figure 7 presents time series of particle size distribution contours with factor intensities and [O$_3$] or [NO$_x$] for the three example time ranges. It can be observed from Fig. 7a,b that F8(morning-N-HOMs), firstly, increases simultaneously with [O$_3$] and, secondly, transforms to F7(day-N-HOMs) with increased NetRad levels. NPF events seem to occur when the intensity of F7(day-N-HOMs) is high enough but it is not simply controlling the strength of a NPF event (defined as the duration of continuing new particle formation during a NPF event). For example in Fig. 7b, the daytime intensity of F7(day-N-HOMs) has an increasing trend beginning on May 12th but the NPF event strength has a decreasing trend beginning already on May 14th. Notable is that also other factors, especially F4(OOMs), have increasing trends during that time range. Therefore, when investigating the fraction $f_{F7}$, it can be seen that it has a decreasing trend beginning on May 14th as well. For the whole studied three-month time range, $f_{F7}$ seems to correlate well with the existence or the strength of a NPF event. There seems to be a critical value for $f_{F7}$ (near 0.5), exceeding of which induces a full NPF event. Additionally, when $f_{F7}$ decreases below the critical value during a NPF event, the event is usually terminated. The diurnal average of $f_{F7}$ is 0.31 on the days showing clear features of NPF events and 0.10 on the days showing hardly any features of NPF events. If considering only a typical time range of observed NPF events, from 10:00 to 20:00, the average of $f_{F7}$ on the days with NPF events is 0.46. In conclusion, other factors than F7(day-N-HOMs) seem to act as an inhibitor for a NPF event, similarly to CS; however, the mechanism behind disfavoring NPF events by the sum of other factors is unknown. Notable is that $f_{F7}$ does not include any particle variables even though particles should act as a sink for particle forming vapors and thus disfavor NPF events. This could be explained partly by the fact that CS and $f_{F7}$ are negatively correlated ($R = -0.30$).

Particle spikes during the harvest time in Fig. 7c occur simultaneously with the F3(nitrophenol, nitrocatechol) intensity and [NO$_x$]. As discussed before in Sect. 4.1, the spikes may be related to tractor emissions. The spikes in [NO$_x$] support the hypothesis of these particles being emitted by tractors but F3(nitrophenol, nitrocatechol) consists mostly of compounds related to vegetation and biomass burning; thus, it is not obvious why it spikes too. The reasons could be that fossil fuel-combusting vehicles emit some compounds found in F3(nitrophenol, nitrocatechol) or that they are released from cut grass during the harvesting process. As already seen from Fig. 6b, increased F3(nitrophenol, nitrocatechol) intensity is connected to increased particle concentrations below 20 nm but not to growing them toward larger sizes and thus not to inducing a NPF event.

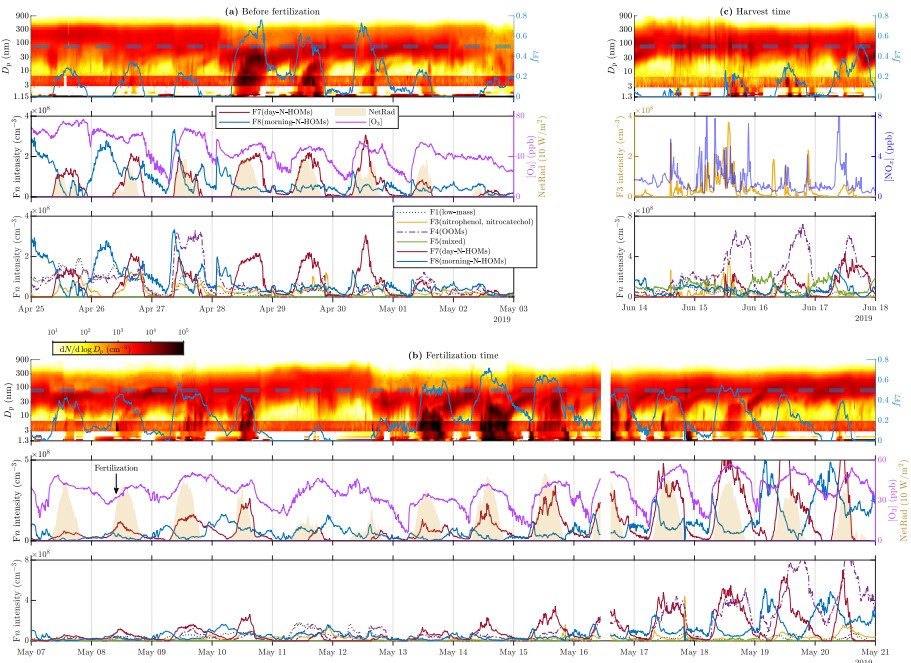

**Figure 7.** Time series of particle size distribution contours and $f_{F7}$ (top panels), F7(day-N-HOMs) and F8(morning-N-HOMs) intensities, [$O_3$], NetRad (shaded areas in the middle panels), and other factor intensities except for the water cluster-based ones (bottom panels) for different example time ranges, **(a)** before the fertilization, **(b)** during and right after the fertilization, and **(c)** during the harvest. Note that, in **(c)**, the middle panel presents F3(nitrophenol, nitrocatechol) intensity and [$NO_x$] instead. The dashed lines shown in the top panels denote the critical $f_{F7}$ value of ∼0.5. Particle data below 3 nm are from a PSM, over 6 nm are from the DMPS, and the remaining part between them are from the NAIS. Note that the PSM data shown in **(a)** are from the PSM,A whereas the data shown in **(b)** and **(c)** are from the PSM,B due to different coverage of the devices. Also note different scales of the y-axes between the subfigures.

It is evident that F7(day-N-HOMs) is connected to particle formation and growth process and F8(morning-N-HOMs) acts as a precursor for F7(day-N-HOMs). The mean intensity of F8(morning-N-HOMs) is 2-fold the mean intensity of F7(day-N-HOMs) with low radiation levels (NetRad $< 130\,\mathrm{W\,m^{-2}}$); conversely, the mean intensity of F7(day-N-HOMs) is 3-fold the mean intensity of F8(morning-N-HOMs) with high radiation levels. This confirms the role of radiation in transforming N-
HOMs from the F8-like form into the F7-like form. The difference in their profiles is further examined in Fig. 8. It can be seen that the clearest differences are, e.g., more $C_4H_6O_4$ (possibly succinic acid) and $C_5H_6O_5$ (possibly ketoglutaric acid) existent in F7(day-N-HOMs) and more $C_7H_4O_6$ (possibly chelidonic acid), $C_7H_{10}O_4$, and $C_9H_{15}O_6N$ existent in F8(morning-N-HOMs). In general, the compounds existing more in F8(morning-N-HOMs) have oxygen-to-carbon (O:C) ratios of around 0.7, but the ones in F7(day-N-HOMs) have O:C ratios of around 0.9, indicating that solar radiation transforming F8(morning-N-HOMs) to F7(day-N-HOMs) is related to oxidation of organic compounds.

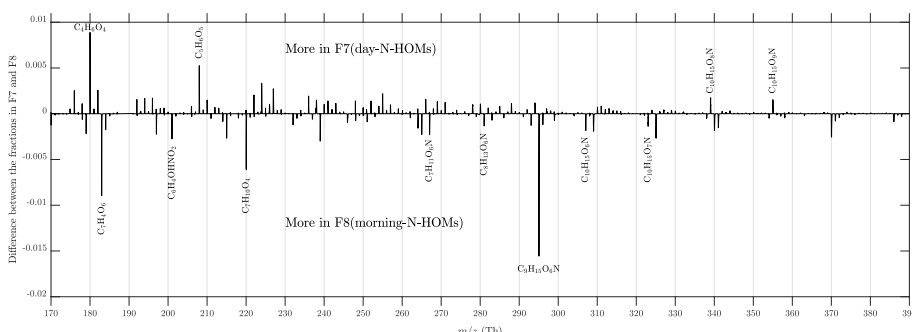

**Figure 8.** Difference between the factors 7 and 8, i.e., the fractions in the F7(day-N-HOMs) profile subtracted by the fractions in the F8(morning-N-HOMs) profile. Identified peaks with the most clear differences between the factor profiles are also presented.

### 4.5   Suggested explanation for particle formation

Figure 9 presents a suggested explanation for particle formation and growth observed in the studied area. The green arrows denote the route leading to a NPF event, which requires ozone to form F8(morning-N-HOMs) which then transforms to F7(day-N-HOMs) with available solar radiation. Finally, the fraction $f_{F7}$ needs to be high enough ($> 0.5$) for a NPF event to be

induced.

Because the occurrence of NPF events seems to be controlled by $f_{F7}$, other factors than F7(day-N-HOMs) can inhibit NPF events if their intensities increase. Additionally, increasing temperature (or its exponential form) leads to more F4(OOMs), which does not favor a NPF event but more likely disfavors it. Hence, radiation is connected to inducing NPF events but temperature to disfavoring them. These are typically connected also to each other but their different roles in NPF may explain

why NPF events are generally observed in all seasons, with high radiation levels but with high $T$ in summers and vice versa in winters. Other variables (F1(low-mass), F5(mixed), F8(morning-N-HOMs), $[HIO_3]$, and $[CH_4SO_3]$) were observed to not induce a NPF event; instead, F1(low-mass), F5(mixed), and F8(morning-N-HOMs) are slightly connected to lowering $f_{F7}$.

As demonstrated in Figs. 1c and 7c, spikes in $[NO_x]$, $[H_2SO_4]$, and F3(nitrophenol, nitrocatechol) lead to particle spikes at the diameters of 20–200 nm, probably due to nearby tractor emissions. The particle spikes are, however, different from NPF

events in terms of their time scales and particle sizes. Figure 6 suggests that $[NH_3]$, $[H_2SO_4]$, and F3(nitrophenol, nitrocatechol) can form sub-10 nm particles but not larger. In theory, there is a possibility that the smallest particles are, in some cases, formed via those but their growth occurs via F7(day-N-HOMs), and a NPF will eventually be observed. Also possible is that particles are both formed and grown via F7(day-N-HOMs). It should be noted that the studied farm may not be the actual source of particle forming precursors as the site is close to the sea and forest as well and that F7(day-N-HOMs) may not include the

actual compound(s) behind the particle formation and growth but is observed simultaneously with the actual compound(s) instead.





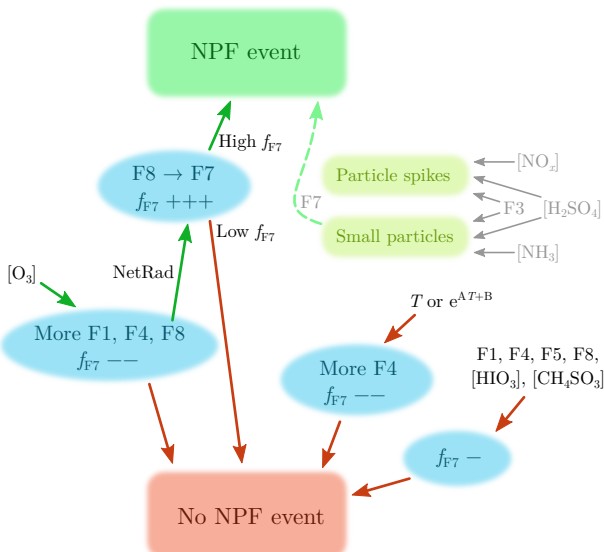

**Figure 9.** Suggested explanation for particle formation and growth observed in the studied area. The green arrows, including ozone, radiation, and high $f_{F7}$, denote the route leading to a NPF event. The red arrows present which variables decrease the probability of a NPF event. The $+$ and $-$ signs after $f_{F7}$ denote the direction and the strength of the change of $f_{F7}$, which needs to be high ($> 0.5$) for a NPF event to be induced. Additionally, increased $[NO_x]$, $[H_2SO_4]$, and F3(nitrophenol, nitrocatechol) levels can be seen as particle spikes at the diameters of 20–200 nm. Moreover, increased $[NH_3]$, $[H_2SO_4]$, and F3(nitrophenol, nitrocatechol) levels can lead to sub-10 nm particles, which can be grown larger with the assistance of F7(day-N-HOMs).

### 4.6 Applications in other studies with nitrate CI-APi-TOF spectra

Estimating the binPMF factors extracted from this study in analyses of other studies can be done without performing a binPMF analysis by using unit mass resolution (UMR) tracers. Table S3 presents Pearson's correlation coefficients between the most

important binPMF factors and UMR data. UMR data are simply the sums from the spectra between, e.g., N-0.5 Th and N+0.5 Th. However, UMR data can lack important information on multiple compounds overlapping at the same UMR mass-to-charge ratio.

For example, one can estimate how F7(day-N-HOMs) would behave in another study by the time series of the spectra between 284.5 Th and 285.5 Th (UMR285). Pearson's $R$ between UMR285 and F7(day-N-HOMs) is 0.91, whereas between

UMR285 and F8(morning-N-HOMs) $R$ is only 0.36 (for comparison, $R = 0.25$ between F7(day-N-HOMs) and F8(morning-N-HOMs)) and between UMR285 and F4(OOMs) $R$ is only 0.65 ($R = 0.47$ between F7(day-N-HOMs) and F4(OOMs)). The most selective UMR tracer would be one that has $R = 1$ for the factor in question but near the ones in Fig. 5 for the all other factors. However, this is not the case for any factor, but the ones in Table S3 denote the most promising ones with their correlation coefficients. If one needs to estimate the variable $f_{F7}$, the most promising tracers are UMR271 divided by





UMR260 ($R = 0.86$ between the division and $f_{F7}$), UMR285 divided by UMR197 ($R = 0.84$), and UMR313 divided by UMR260 ($R = 0.84$). However, it should be noted that the critical value depends on the selected UMR tracers.

## 5    Conclusions

New particle formation events occurring in a coastal agricultural site were examined by performing measurements of gases, molecular clusters, particles, and environmental parameters. The area is a pilot agricultural farm for regenerative farming, Qvidja, located in Southwestern Finland. This study covers roughly three months of the measurement data recorded between April and June in 2019, when a laboratory van, ATMo-Lab, was parked next to a stationary measurement container located in the middle of the farming fields. The CI-APi-TOF mass spectrometer was used to measure potential new particle-forming acids, sulfuric acid, iodic acid, and methanesulfonic acid, together with a multitude of other compounds. In addition to these acids, the high resolution-mass spectra between 169 and 450 Th were elaborated via the binned positive matrix factorization (binPMF) method. Eight binPMF factors were selected as the optimal set of factors.

From all 84 measurement days, 25 days showed clear features of NPF events. The NPF event days can be partly explained with ammonia, sulfuric acid, and condensation sink levels. The concentrations of ammonia and sulfuric acid were generally higher and condensation sink lower during the NPF events, but still without a clear formula on the occurrence of the events. Iodic acid and methanesulfonic acid were observed to not correlate with the NPF events. An even better explanation is, instead, achieved when examining the levels of the binPMF factors. It was observed that the factor F7(day-N-HOMs) is high during the NPF events. Further examination shows that the events can be explained very accurately using a single variable, $f_{F7}$, which denotes the fraction of the spectra between 169 and 450 Th that is in the form of F7(day-N-HOMs). In all NPF events, $f_{F7}$ exceeded a critical value of $\sim 0.5$ and the total time of the exceeding corresponds to the length of the event. Surprisingly, no particle values are needed to predict NPF events in this case even though the condensation sink generally disfavors the events.

Pearson's correlation coefficient between F7(day-N-HOMs) and particle concentrations at every measured particle size bin is positive from near 1 nm up to 300 nm, but all other factors behave differently. F3(nitrophenol, nitrocatechol) is positive for the smallest particle sizes but approaches zero near 20 nm, which can be interpreted so that F3(nitrophenol, nitrocatechol) may be involved in NPF but not in growing them past the size of 20 nm and thus not inducing NPF events. All other factors, instead, have negative correlation coefficients for particles smaller than 40 nm. F3(nitrophenol, nitrocatechol) together with sulfuric acid and nitrogen oxides was also connected to the appearance of short particle spikes in the size range of 20–200 nm, which are presumably due to the emissions of tractors harvesting the fields.

Examination of time series revealed that the intensity of F7(day-N-HOMs) is elevated with solar radiation when F8(morning-N-HOMs) exists, like F8(morning-N-HOMs) is transformed to F7(day-N-HOMs). Instead, F8(morning-N-HOMs) seems to be formed through ozonolysis because its intensity is elevated simultaneously with the ozone concentration. In conclusion, NPF events observed at the studied coastal agricultural environment follow this routing: ozone levels elevate which causes F8(morning-N-HOMs) intensity to elevate, which is then transformed to F7(day-N-HOMs) via radiation; if F7(day-N-HOMs) is the major form in the spectra, a NPF event is observed.



Investigation of the high resolution-spectra of F7(day-N-HOMs) and F8(morning-N-HOMs) shows that they both consist mainly of HOMs with an extra nitrogen atom, but the compounds existing more in F7(day-N-HOMs) have higher oxygen-to-carbon ratios than the ones in F8(morning-N-HOMs). Additionally, there seem to be compounds related to the citric acid cycle in F7(day-N-HOMs) but not in F8(morning-N-HOMs). Notable compounds in their spectra are $C_{10}H_{15}O_8N$ (339.0585 Th) in F7(day-N-HOMs) and $C_9H_{15}O_6N$ (295.0749 Th) in F8(morning-N-HOMs). $C_{10}H_{15}O_8N$ is the compound with which the first indication of the connection between HOMs and atmospheric nucleation was observed (Kulmala et al., 2013). Another factor containing HOMs was also obtained from the binPMF analysis, F4(OOMs). It was observed to not favor NPF events, but more likely to disfavor instead, due to its elevated intensity with higher temperatures causing lower $f_{F7}$.

In estimating the behavior of F7(day-N-HOMs) and F8(morning-N-HOMs) in any other study with a nitrate ion-based mass spectrometer without performing the binPMF analysis, time series of those compounds could be useful in estimating. If only unit mass resolution-data is available, the time series of 220 or 236 Th could be examined in estimating F4(OOMs), of 285, 271, or 339 Th in estimating F7(day-N-HOMs), and of 295 or 265 Th in estimating F8(morning-N-HOMs). The time series of 271 Th divided by the time series of 260 Th is the best estimate for examining the (relative) behavior of the fraction $f_{F7}$.

*Data availability.* Time series data measured at the Qvidja farm from Apr to Jun 2019 together with the raw spectra of the binPMF factors are freely available at https://doi.org/10.5281/zenodo.6394454 (Olin, 2022).

*Author contributions.* MDM, MS, TP, and MS designed the research. MOl, MOk, JK, ML, YW, and JD performed the measurements. MOl and MOk handled the measurement data. MOl performed the PMF analysis and prepared the paper with contributions from all co-authors.

*Competing interests.* Some authors are members of the editorial board of Atmospheric Chemistry and Physics. The peer-review process was guided by an independent editor, and the authors have also no other competing interests to declare.

*Acknowledgements.* University of Helsinki and Qvidja technical staff and Qvidja farm owners are acknowledged for their support on the site. We thank the tofTools team for providing tools for mass spectrometry analysis. This research has been supported by the Academy of Finland through the Condenz project (grant nos. 310627 & 326437), through the ACCC Flagship grant (no. 337551), and through the Academy Research Fellow grant (no. 331207).



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
