# Peer review of "Measurement report: Atmospheric new particle formation in a coastal agricultural site explained with binPMF analysis of nitrate CI-APi-TOF spectra"

_Atmospheric Chemistry and Physics, 2022_

## Referee Comment (RC2)

**General comments:**

In this paper, the authors investigated the new particle formation (NPF) events in a coastal agricultural site in Southwestern Finland by using a combination of a nitrate ion-based chemical-ionization mass spectrometer, gas analyzers as well as aerosol samplers. The binned positive matrix factorization method (binPMF) was applied to the measured mass spectra, showing that eight factors could describe the time series of ambient gas and cluster composition during the NPF events. Before publication, I think there are several comments that the authors may need to consider.

1.  There are several uncertainties in this study which may lead to some problems or make this study not really convincing. First, the mass errors ranged from -10 ppm to 50 ppm (Line 211), so the identification of compounds with a high molecular weight may be not correct. How did the authors determine the confidence levels of the identifications in Table 1? Second, the authors said that "it cannot be certainly proved that a variable is actually forming new particles or growing them by examining the correlations. There is always a possibility that a variable is only observed simultaneously with NPF events due to the similarity of its source and the source of the precursor really causing the NPF events." I agree with the authors about this point, but does it also mean the results of this study are also based on this uncertainty?

2.  What do the F8 compounds come from? I also think the authors need to give a map showing the sampling site and the meteorological information such as the wind speed and direction is also needed to illustrate the sources of measured aerosols and gases.

3.  The time profiles of F7 compounds did not correlate with F8 compounds (Figure 7), I do not understand why the F7 formed from the F8?

4.  Did the authors detect halogenated organics due to the proximity of the measurement site to the sea?

**Specific comments:**

1. Line 9: "Values of $f_{F7}$ higher than 0.5 were typically observed during the NPF events". However, Figure 7c showed this value is lower than 0.5 during the NPF events on May 8-11 and 17.

2. Line 75: What is the mass resolution of CIMS during the field observation?

3. Line 108: "Ions smaller than 169 Th were omitted because there are many organic compounds that are unlikely the key compounds in NPF". However, methanesulfonic acid can also efficiently initiate NPF in the presence of small alkylamines and water (Chen et al., 2016; Dawson et al., 2012).

**References:**

Chen, H., Varner, M. E., Gerber, R. B. and Finlayson-Pitts, B. J.: Reactions of Methanesulfonic Acid with Amines and Ammonia as a Source of New Particles in Air, J. Phys. Chem. B, 120(8), 1526–1536, doi:10.1021/acs.jpcb.5b07433, 2016.

Dawson, M. L., Varner, M. E., Perraud, V., Ezell, M. J., Gerber, R. B. and Finlayson-Pitts, B. J.: Simplified mechanism for new particle formation from methanesulfonic acid, amines, and water via experiments and ab initio calculations, Proc. Natl. Acad. Sci. U. S. A., 109(46), 18719–18724, doi:10.1073/pnas.1211878109, 2012.

---

## Author Comment (AC1)

**Final response to the referees' comments for Olin et al.: "Measurement report: Atmospheric new particle formation in a coastal agricultural site explained with binPMF analysis of nitrate CI-APi-TOF spectra"**

We thank the referees for their very useful comments and have corrected the manuscript according to all of them.

Referee reports are in *black italic* and authors' responses in blue roman font. **Bold blue** or  fonts highlight changed text parts in some comments. The marked-up manuscript and Supplement highlighting the changes are included at the end of this file.

**Referee comment #1:**

*General comment:*

*In this measurement report, "Measurement report: Atmospheric new particle formation in a coastal agricultural site explained with binPMF analysis of nitrate CI-APi-TOF spectra" Olin et al. performed nitrate-CIMS measurement at a coastal site. BinPMF was applied to the CIMS data, reporting two factors- F7 and F8, with high mass-to-charge ratios factor, which explained the NPF events well. It is an interesting study, and the results appear to support the conclusion. I have some minor comments the authors may want to consider before publication. My biggest concern is the uncertainties of the measurement as it can lead to some problems when comparing the intensities of different factors.*

*Specifics:*

**1.** *Abstract. You may want to associate F7 or F8 with a more physically meaningful name, as it is hard to comprehend what is the chemistry behind F7/F8.*

We agree that the text in the abstract was not clear enough to understand the chemistry behind the factors 7 and 8, which was actually not very deeply examined in this measurement report. Therefore, we decided to keep their original names, *F7(day-N-HOMs)* and *F8(morning-N-HOMs)*, because they only denote how the factors appeared in the data, which can be provided with much higher certainty than any chemistry-related names with the level of analysis in this measurement report. We also didn't want to mention their names already in the abstract; instead, the text in the abstract is now clarified. The sentence near the middle part of the abstract is now updated to "... both having profiles with patterns of highly oxygenated organic molecules **with one nitrogen atom**." and the last sentence updated to "It was also observed that the factor 8 acts  **like a precursor for** the factor 7 with  **solar radiation and that the formation of the factor 8 is associated with ozone** levels." The change of wording from "as a precursor" to "like a precursor" is related to the referee #1 comment 19 and the referee #2 comment 3.

**2.** *Line 73. Provide more details about how the instrument works in principle. E.g., how the agent ion was produced, and at what rate? What is the flow rate being sampled into the CIMS?*

Descriptions of the generation of the reagent ion and of the flows of the CI-inlet are now added to the text: "$NO_3^-$ **was produced by adding nitric acid ($HNO_3$) vapor to the sheath air which was then exposed to X-ray radiation inside the CI-inlet. The $HNO_3$ vapor was produced by directing 10 sccm flow of the sheath air through a glass vial containing liquid $HNO_3$. The sheath air was a 20 slpm flow of outdoor air filtered using an activated carbon filter and a HEPA filter.** The **outdoor** air sample was drawn **to the CI-inlet** with the flow rate of 8.3 slpm through a side hatch of the van with a 16 mm pipe of 60 cm length**, which also acts as a laminarization tube for the CI-inlet**."

**3.** *Line 79. Any specific reason for averaging to 600 s? Is the code open-sourced, available at what site?*

The averaging time needs to be long enough to have an adequate signal-to-noise ratio but shorter than the timescale of expected variations in the signal. At this agricultural environment, no rapid variations in the signal are expected and 600 s is a best estimate for the suitable averaging time in this case. This is now clarified in the text with the updated sentence "The data were recorded in 4 s time resolution but were first averaged to 600 s resolution ($t_a$) **in order to increase signal-to-noise ratios (SNRs) but still maintain an adequate time resolution for the campaign**."

The tofTools code is not fully open-sourced and it has been developed by many authors from many institutions. It has formerly been openly available, but its homepage has not been operated anymore. Therefore, no download link cannot be given for it, but the latest codes may be obtained by contacting the main developer, Prof. Heikki Junninen from University of Tartu.

**4.** *Line 80. What are the errors for the m/z calibration?*

The errors of the mass calibration during the campaign were 0.4 ppm, in median. However, there were eight nights near the end of the campaign with the errors up to 80 ppm. This was caused by the high signals from $H_2CO_3$ causing overlapping peaks to the $NO_3^-$ and $NO_3^- \cdot HNO_3$ peaks used in the calibration. Nevertheless, that didn't actually cause any excessive errors to the result of the mass calibration, since the mass calibration parameters did not include any significant deviation during those nights. Only the choice of the peaks for the calibration was not very optimal for those nights. These values and this discussion is now included in the text (in the results part, in Sect. 4.2.2) with the added text of "**The median error of the $m/z$ calibration during the campaign was 0.4 ppm. However, the errors were as high as 80 ppm during eight nights near the end of the campaign due to high $H_2CO_3$ signals causing overlapping signals to the $NO_3^-$ and $NO_3^- \cdot HNO_3$ peaks used in the calibration. Nevertheless, the result of the calibration, which is the parameters used to convert time-of-flight to $m/z$, did not include any significant deviation during those nights, implying that the calibration itself was successful although the choice of the peaks for the calibration were not optimal for those nights.**"

**5.** *Line 85-90. I assume the C value is obtained by your calibration, while P is from the literature? Just wondering if the C and P values would vary a lot over different times and with different instruments.*

The $C$ value is obtained from three calibration runs, performed once a month at the measurement site. The highest obtained $C$ was 31% higher than the lowest one. Therefore, the variation in $C$ over different times is not very high. This level of variation is now mentioned in the text by expressing the value of $C$ in the form of "$(2.45 \pm 0.33) \times 10^9 \, \mathrm{cm}^{-3}$" instead of only "$2.45 \times 10^9 \, \mathrm{cm}^{-3}$". Additionally, "**(once a month)**" is now mentioned in the text.

The $P$ value is obtained by calculating the diffusional losses of $H_2SO_4$ inside a 60 cm long tube having the flow rate of 8.3 slpm using the function by Gormley and Kennedy (1948). The obtained penetration efficiency of 0.58 includes a minor uncertainty of 0.03, which results from the uncertainties in the measurements of the sampling pipe length and of the flow rate and from the variation in relative humidity (RH) of the outdoor air, since the diffusion coefficient of $H_2SO_4$ is effectively dependent on RH due to hydration of the $H_2SO_4$ molecules. This information is now included in the text with the updated sentence "$P$ is the penetration efficiency of $H_2SO_4$ in the  **60 cm long sampling pipe having the flow rate of 8.3 slpm** ($= 0.58 \pm 0.03$**) determined with the function by Gormley and Kennedy (1948)**".

**6.** *Line 90. I am not familiar with nitrate CIMS calibration, can you explicitly explain what "penetration efficiency" is?*

The 60 cm long sampling pipe is used to make the flow to the CI-inlet laminar, required for its optimal operation. The length of the sampling pipe and the flow rate in it vary between campaigns, but the calibration source (producing a known $H_2SO_4$ concentration) is typically placed right at the front of the CI-inlet, i.e., downstream of the sampling pipe. Therefore, $C$ is used to convert the signal to the $H_2SO_4$ concentration at the front of the CI-inlet. The concentration at the front of the sampling pipe is, however, different because a part of the $H_2SO_4$ molecules are lost onto the inner walls of the sampling tube via diffusion. With the used setup, $P$ was 0.58, denoting that 58% of $H_2SO_4$ molecules penetrates the sampling tube while 42% are lost onto the walls. This is now clarified in the text with the updated sentence mentioned with the previous comment. Some studies report $\frac{C}{P}$ as a corrected calibration coefficient, which would be $4.22 \times 10^9 \, \mathrm{cm}^{-3}$ in this study.

**7.** *Line 105. Elaborate on the uncertainties for the determination of H2SO4, HIO3, and CH4SO3.*

The uncertainty in the $H_2SO_4$ concentration is caused by the uncertainties in $C$ and $P$, resulting in the relative uncertainty in the $H_2SO_4$ concentration of 15 %, when the uncertainty in $C$ is determined with its variation between three caribration runs during the campaing. There is, however, a systematic uncertainty in $C$ involved in a single calibration run (Kürten et al., 2012) but it is not easily quantified. A frequently used rough approximation for the uncertainty in the determined $H_2SO_4$ concentrations is $-50\%/+100\%$. This discussion is now included with the added text "**[H$_2$SO$_4$] has a relative uncertainty of 15 %, originating mostly from the uncertainty in $C$, determined with its variation between three calibration runs during the campaign. However, because a single calibration run also involves its own systematic uncertainty (Kürten et al., 2012)—which is not easily quantified—the uncertainty in [H$_2$SO$_4$] can be higher (a frequently used rough approximation is $-50\%/+100\%$).**"

The uncertainties in the concentrations of $HIO_3$ and $CH_4SO_3$ during the campaign are basically also 15 % (or even $-50\%/+100\%$), similarly to $H_2SO_4$, but their absolute levels can, however, be different in reality because $C$ and $P$ have not been determined for them. Nevertheless, the absolute levels do not affect the results of this study, since the differences between $C$ and $P$ for different compounds are expected to be constant over different times. Additionally, in the case of $HIO_3$ and $CH_4SO_3$, $C$ values for them are also expected to be near the $C$ value for $H_2SO_4$ because they all have collision-limited charging efficiency when reacting with $NO_3^-$ ions (Simon et al., 2020; Beck et al., 2021; Wang et al., 2021). The $P$ value is also not highly dependent on the compound within the range of compounds in question in this study. This discussion is now included in the text with the sentences "**Uncertainties in [HIO$_3$] and in [CH$_4$SO$_3$] are not easily quantified but the relative uncertainty of 15 % in [H$_2$SO$_4$] (or $-50\%/+100\%$) can be used in estimating relative uncertainties during the campaign. However, the absolute levels of [HIO$_3$] and [CH$_4$SO$_3$] can still differ more due to the approximation of using $C$ and $P$ determined for H$_2$SO$_4$, but the difference between the calibration coefficients for different compounds is expected to be nearly constant during the campaign.**"

**8.** *Line 108, "Ions smaller than 169 Th were omitted because there are many organic compounds that are unlikely the key compounds in NPF and have relatively high signals possibly causing issues in the binPMF analysis" It is not clear why many organic compounds are unlikely the key compounds in NPF.*

This is now clarified in the text with the added sentences "**Compounds with lower masses typically have higher volatilities and are thus more unlikely participating in nucleation and condensation. However, it is not generally true since, e.g., H$_2$SO$_4$ and CH$_4$SO$_3$ are below 169 Th but still known to contribute to NPF. Therefore, they were analyzed separately of the binPMF analysis.**"

**9.** *Line 121. It is not clear why X needs to be normalized and how it is normalized? Is the signal used to calculate H2SO4 concentrations normalized?*

The signal used to calculate the $H_2SO_4$ concentration is normalized with the total signal of nitrate ions. The normalization is done with the denominator in Eq. (1). This is needed because signals have always to be expressed relative to the reagent ion signal, which is not very constant due to changes in temperature and the level of liquid $HNO_3$ in the glass vial. The normalized signal is a dimensionless variable, which is then converted to the concentration-based unit by multiplying it with a calibration coefficient. Similarly to the $H_2SO_4$ concentration, the data matrix X needs to be normalized with the reagent ion signal in order to prevent variations in the reagent ion signal to result in erroneous variations in the data of interest. This way, the data matrix X will also be expressed dimensionlessly, and the result of the binPMF run, the time series matrix TS, can also be expressed in a concentration-based unit (after multiplying it with $\frac{C}{P}$). This information is now included in the text with the updated sentences "The data matrix X was normalized before running the binPMF code with the reagent ion signals as **is done with the denominator** in Eq. (1) for calculating **[H$_2$SO$_4$]. This way the data matrix becomes dimensionless and expressed relative to the NO$_3^-$ concentration.**"

The uncertainty matrix S needs to be expressed in the same unit as X and is thus normalized also with the nitrate ion signal. This is also now mentioned in the text with the updated sentence "Similar to the matrix X, the matrix S was also normalized with the reagent ion signals before running the binPMF code **to have them expressed with the same units (dimensionless).**"

**10.** *Line 125. I am slightly confused as it seems Sij is associated with moving median, but in equation (4) no median values were used. Is the a value of 1.35 a reference value from literature? Or determined by what method specifically.*

We agree that the construction of the uncertainty matrix S was not described clearly enough. Moving median values were used to determine the values for $a$ and $\sigma_{\text{noise}}$. This was done by, first, determining uncertainties of the signals for all time bins and $m/z$ bins separately using the deviations between the signals and their moving medians. Then, the obtained uncertainties were fitted to the Poisson distribution for all $m/z$ bins separately but using all time bins. This results in Eq. (4) where $a$ and $\sigma_{\text{noise}}$ are functions of the $m/z$ bin. Finally, weighted means of $a$ and $\sigma_{\text{noise}}$ over all $m/z$ bins are calculated, which leads to their mentioned values with uncertainties ($a = 1.35 \pm 0.22$ and $\sigma_{\text{noise}} = (0.001 \pm 0.003)\,\text{cps}$). This method is described in more detail in the study by Yan et al. (2016). The paragraph related to this is now rewritten and it now reads "The binPMF code tries to find the matrices TS and MS producing the lowest possible value for the sum of the scaled residuals,

$$Q = \sum_{i=1}^{9970} \sum_{j=1}^{6975} (\text{R}_{ij}/\text{S}_{ij})^2, \tag{3}$$

where S is the uncertainty matrix. S was estimated from the ambient data via the method suggested by Yan et al. (2016), i.e., through approximating instrument noise as the difference between the measured signal and its moving median over 5 data points. **The differences are determined separately for all 9970 time bins and for all 6975 $m/z$ bins.** Assuming that the counting statistics follow the Poisson distribution, the function

$$\text{S}_{ij} = \sigma_{ij} + \sigma_{\text{noise}} = a \cdot \sqrt{\frac{\max\left(I, \frac{1}{t_a}\right)}{t_a}} + \sigma_{\text{noise}} \tag{4}$$

where $a$ is the correcting factor incorporating any unaccounted contributions to the uncertainty (Allan et al., 2003) and $I$ is the signal intensity in counts per second (cps), **can be described to express the instrument noise. The values for $a$ and $\sigma_{\text{noise}}$ were first determined for all $m/z$ bins separately by fitting the differences between the measured signal and the moving median for different levels of $I$ using all time bins and Eq. (4). The obtained $a$ and $\sigma_{\text{noise}}$ values were then averaged over all $m/z$ bins using the successes of the fittings as weighting factors. Finally, the values of $a = 1.35 \pm 0.22$ and $\sigma_{\text{noise}} = (0.001 \pm 0.003)\,\text{cps}$ were obtained, and the function for calculating the uncertainty matrix became**

$$\text{S}_{ij} = 1.35 \cdot \sqrt{\frac{\max\left(I, \frac{1}{600\,\text{s}}\right)}{600\,\text{s}}} + 0.001\,\text{cps}. \tag{5}$$

The value of $a$ **obtained here**, $1.35 \pm 0.22$, is on a similar level to the values in the study by Yan et al. (2016), $1.28 \pm 0.09$ or $1.1 \pm 0.3$. Similar to the matrix X, the matrix S was also normalized with the reagent ion signals before running the binPMF code **to have them expressed with the same units (dimensionless)."**

**11.** *Line 137. Define "substantially". A low Q/Qexp is not necessarily the best PMF solution.*

This sentence is now updated, to better understand how $p$ was changed, to "The code was run with increasing $p$ until the ratio $Q/Q_{\text{exp}}$  **set to a level with no further decrease**."

In this study, the optimal $p$ value of 8 was selected by considering the behavior of the $Q/Q_{\text{exp}}$ ratios with different values of $p$, by minimizing $p$ to keep the analysis as simple as possible, and by considering the number of factors having organic patterns. This didn't lead to the lowest $Q/Q_{\text{exp}}$ ratio but to near that. Additionally, because the results of PMF analyses, in general, include rotational ambiguity, we added the following sentences to the text: "**The lowest $Q/Q_{\text{exp}}$ is not necessarily the best solution of PMF because the solution of PMF is not unique due to the nature of PMF. For example, due to its rotational ambiguity, there are basically infinite number of solutions providing equal $Q$ values (Paatero et al., 2002). Rotational ambiguity is, however, not considered in this study because it is typically not a problem with ambient measurement data, which usually contain enough data points with values near zero (Zhang et al., 2019)."**

**12.** *Line 145. Again, what is the uncertainty for the assumptions of similar calibration coefficient and penetration efficiency?*

This is now discussed with the added text "**Penetration efficiencies are not expected to differ notably between $H_2SO_4$ and the compounds with $m/z$ of 169–450 Th, detected by nitrate CI-APi-TOF, but calibration coefficients are. $C$ for $H_2SO_4$ denotes a value near the highest possible sensitivity, since the charging efficiency of $H_2SO_4$ is collision-limited (Simon et al., 2020). However, $C$ can be even orders of magnitude higher for compounds with weaker binding energies with $NO_3^-$ (Hyttinen et al., 2015). Therefore, the sensitivity of the used system can be much lower for some compounds. This is true especially for organic compounds with low oxygen-to-carbon (O:C) ratios, since organic compounds with adequately large number of carbon atoms having O:C$\geq$0.6 are estimated to have similar collision-limited charging efficiencies to $H_2SO_4$ (Simon et al., 2020), but the binPMF factors of this study includes compounds with lower O:C ratios too. As it is practically impossible to determine calibration coefficients for all detectable compounds, this commonly used approximation is used here only to convert the dimensionless variables in TS to a practical concentration-based unit.**"

**13.** *Line 155. Define "strong" and "weaker". Do you mean particle number concentration?*

The definition of the strength of a NPF event, "(the strength of a NPF event is defined as the duration of continuing NPF during a NPF event)", was mentioned in Sect. 4.4 but is now moved already here to Sect. 4.1.

**14.** *Line 157. How was CS determined? Maybe explain it in the Method section.*

CS was calculated from the DMPS data with the method by Kulmala et al. (2012) using a common approximation of assuming $H_2SO_4$ as the condensing vapor. This is now mentioned already in the "Measurement instruments" section with the updated sentence "DMPS was used to determine the particle size distribution in the size range of 6–823 nm **and CS, which was calculated with the method by Kulmala et al. (2012) using a common approximation of assuming $H_2SO_4$ as the condensing vapor.**"

**15.** *Line 175. It is hard to comprehend this sentence. Define "smallest particles" and rephrase "simultaneously not elevated"*

This sentence is now clarified to "They originate probably from direct emission sources, e.g., from tractors harvesting the fields, rather than via NPF because  **there is no simultaneous concentration increase in particles smaller than those**."

**16.** *Line 193. Any data to support the statement of "(3 factor profiles having organic patterns with p = 6 or p = 7)".*

The asked supporting data is now added to the Supplement as Fig. S1 and the text in question is now updated to "( **three** factor profiles having organic patterns with $p = 6$ or $p = 7$, **see Fig. S1**)".

[Figure]

**Figure S1.** Normalized spectra of all binPMF factors if (a) $p = 6$ or (b) $p = 7$ were used. Both of these factor sets have three factor profiles having organic patterns, whereas $p = 8$ results in four factor profiles having organic patterns and is further analyzed in this study.

**17.** *Line 280. Has NetRad been mentioned before?*

Yes, it has been mentioned already in Sect. 2.2 with the text "net radiation (NetRad)".

**18.** *Line 353. F8 and F7 may have different sensitivity in CIMS. You may need to discuss the uncertainties when comparing their intensities.*

Different sensitivities of the compounds detected by the CI-APi-TOF have been discussed already in Sect. 3.2 (related to the referee #1 comment 12). Here, in Sect. 4.4, these possible differences are now considered by updating the text to "**Assuming similar sensitivities of F7(day-N-HOMs) and F8(morning-N-HOMs) in the CI-APi-TOF,** the mean intensity of F8(morning-N-HOMs) is 2-fold the mean intensity of F7(day-N-HOMs) with low radiation levels ($\mathrm{NetRad} < 130\,\mathrm{Wm^{-2}}$); conversely, the mean intensity of F7(day-N-HOMs) is 3-fold the mean intensity of F8(morning-N-HOMs) with high radiation levels."

**19.** *Line 354. I don't see why it confirms the transformation from F8 to F7 as F8 decreases while F7 increases. There are other possibilities e.g., transport of pollutants with different intensities, and changes in air masses.*

It is true that F8 decreasing and F7 increasing with radiation does not necessarily confirm the role of radiation in this process. However, by looking at the time series of F7, F8 and $O_3$ in Fig. 7, it can be observed that when NetRad increases, F8 decreases and almost the same amount is added to F7, while their sum behaves very similar to the trend of the $O_3$ concentration. Therefore, and due to the behavior of F7 and F8 with NetRad levels (see the previous comment), there is clearly something related to radiation and the abundance of F7 and F8. As mentioned by the referee, there are other possibilities, such changes in air masses, causing the behavior of F7 and F8 with radiation levels. The word "confirms" in the sentence in question is now replaced with the word "supports" to clarify that it was not fully confirmed and also the text "**It cannot be proven with these measurements that this transformation is actually a real chemical process since the behavior of the factors with NetRad levels can also be resulted, e.g., from possibly different origins of the air masses between the daytime and the nighttime. Nevertheless, the current knowledge cannot at least exclude the transformation via radiation, because the compounds in F7(day-N-HOMs) are basically more oxidized than the ones in F8(morning-N-HOMs) (discussed later in Sect. 4.4) being physically reasonable since photochemistry typically leads to oxidation.**" is now added to Sect. 4.3. See also the referee #2 comment 3, which is related to this as well.

**20.** *Line 379. "…temperature to disfavoring them…" do you specifically mean high or low temperatures?*

High temperatures were meant. The word "high" is now added.

**21.** *Conclusion. You may want to make the conclusion section short, highlighting the new findings in this study.*

The conclusions section is now shortened from ∼44 lines to ∼31 lines. Discussions about Pearson's correlation coefficients with different particle sizes and about estimating the behavior of F7 and F8 in other studies using UMR data are now removed from there (see the marked-up manuscript for the changes in it).

**22.** *Figure 1. What is the red ban between Dp 3-8 nm?*

The red banners in that particle size range are caused by that the data for this particle size range is obtained from the NAIS, which has a general tendency to overestimate concentrations at those particle sizes (Gagné et al., 2011; Mirme and Mirme, 2013; Kangasluoma et al., 2020). This is now mentioned in the caption of Fig. 1 with the text "**The data from the NAIS show red banners due to its general tendency to overestimate concentrations at these particle sizes (Gagné et al., 2011; Mirme and Mirme, 2013; Kangasluoma et al., 2020).**"

**23.** *Figure7. The label/legends in this figure are small, you may want to make it visible at a font size of at least 8.*

All the texts in Figs. 1 and 7 are now increased. Due to that, the legend for the bottom panels of Fig. 7 was needed to be moved from the (a) plot to the (c) plot; thus, the text "**the legend is shown in the bottom panel of (c)**" was added to the caption of Fig. 7.

**Referee comment #2:**

*General comments:*

*In this paper, the authors investigated the new particle formation (NPF) events in a coastal agricultural site in Southwestern Finland by using a combination of a nitrate ion-based chemical-ionization mass spectrometer, and gas analyzers as well as aerosol samplers. The binned positive matrix factorization method (binPMF) was applied to the measured mass spectra, showing that eight factors could describe the time series of ambient gas and cluster composition during the NPF events. Before publication, I think there are several comments that the authors may need to consider.*

**1.** *There are several uncertainties in this study that may lead to some problems or make this study not really convincing. First, the mass errors ranged from -10 ppm to 50 ppm (Line 211), so the identification of compounds with a high molecular weight maybe not be correct. How did the authors determine the confidence levels of the identifications in Table 1? Second, the authors said that "it cannot be certainly proved that a variable is actually forming new particles or growing them by examining the correlations. There is always a possibility that a variable is only observed simultaneously with NPF events due to the similarity of its source and the source of the precursor really causing the NPF events." I agree with the authors about this point, but does it also mean the results of this study are also based on this uncertainty?*

The mass errors before the binPMF run were 0.4 ppm, in median (see the referee #1 comment 4). However, because the $m/z$ bin width in the binPMF needs to be a finite value, summing all signal within a $m/z$ bin results in losing of the information on how the signal behaves within the bin. In this study, the bin width of 0.02 Th was used, as by Zhang et al. (2019). Choosing narrower bin width would result in better mass resolution but in lower signal-to-noise ratio, because the sum within a bin would be smaller. Therefore, we didn't want to alter the bin width for this study. Based on the mass errors of the water clusters, the mass errors after the binPMF run were between -10 ppm and 0 ppm when $m/z$ is below 280 Th and increase from 0 ppm to +50 ppm when $m/z$ increases from 280 Th to 400 Th. Fortunately, the main interest in the spectra in the binPMF factors is actually quite near 280 Th, for which the mass error is near 0 ppm.

  Additionally, the mass errors approaching +50 ppm for the larger masses is eventually not as problematic as it may sound like. That is because the mass errors are known for the water clusters. These known errors can be used to estimate what the mass errors for any compound with a specific $m/z$ value should be. For example, the peak at 201.0171 Th in F3 is $NO_3^- \cdot C_6H_4OHNO_2$ (nitrophenol) with the mass error of -9.3 ppm, while the nearest water cluster peak, at 197.0278 Th in F2, is $NO_3^- \cdot HNO_3(H_2O)_4$ with the mass error of -7.8 ppm. Because these errors (-9.3 ppm vs. -7.8 ppm) are so close to each other, it can be identified with a high certainty that the peak at 201.0171 Th in F3 is really $NO_3^- \cdot C_6H_4OHNO_2$. Similarly, in a case of a compound with a higher molecular weight, for example the peak at 339.0585 Th in F7 having the error of +28 ppm for $C_{10}H_{15}O_8N$, can also be identified with a quite high certainty although the errors sound large (the mass errors of water cluster peaks near this $m/z$ range are +25 ... +35 ppm).

  This is now clarified in the text, first, with the added text "**aided by observed errors of $m/z$ ratios of water clusters (see Sect. 4.2.3)**" to the sentence "Nevertheless, identifying possible chemical formulae from the observed peaks was still done with relatively high confidence for several peaks." in Sect. 4.2.2 and, secondly, with the added text "**The confidence levels are based on subjective estimations on the success of peak fitting (aided by the errors of water clusters at specific $m/z$ ratios and by the known isotopic patterns) and on the expectation of the compound to be detected with the used instrumentation.**" to the caption of Table 1. This addition answers also to the question about how the confidence levels of the identifications were defined.

  As for the second concern in this referee comment, the uncertainty in the knowledge of the precursor really causing NPF events is always present in similar studies. Therefore, the observed correlations between any variable and NPF should also

be assessed by considering whether they are physically reasonable or not. For the compounds found in F7 leading to NPF events, there is currently no proof against it since it is not implausible that organic compounds with presumably low volatilities could form new particles and grow them. We have used wordings that do not strictly say that, e.g., "F7 causes NPF". Instead, wordings like "It was observed that the factor 7 had elevated intensities during the NPF events." in the abstract and "a suggested explanation for particle formation and growth observed in the studied area" in Sect. 4.5 have been used to emphasize the involved uncertainty.

The sentence "**This kind of uncertainty is, however, present in all similar studies as well; and therefore, the observed correlations should also be assessed by considering whether they are physically reasonable or not.**" is now added after the sentence "There is always a possibility that a variable is only observed simultaneously with NPF events due to the similarity of its source and the source of the precursor really causing the NPF events." in Sect. 4.3. Additionally, there is a sentence in the conclusions section which is now updated to "In conclusion, NPF events observed at the studied coastal agricultural environment **seem to** follow this routing: ozone levels elevate which causes F8(morning-N-HOMs) intensity to elevate, which is then transformed to F7(day-N-HOMs) via radiation; if F7(day-N-HOMs) is the major form in the spectra, a NPF event is observed."

In addition to this discussion on F7 and NPF, some too strict wordings used to describe the transformation of F8 to F7 are also now loosened (see the answers to the referee #1 comment 19 and the referee #2 comment 3).

**2.** *Where do the F8 compounds come from? I also think the authors need to give a map showing the sampling site and the meteorological information such as the wind speed and direction is also required to illustrate the sources of measured aerosols and gases.*

In this measurement report, we do not examine the actual origins of the detected compounds and binPMF factors. The idea here is to report what has been observed but not to deeply examine the root causes of the observations. Therefore, we decided not to examine where the compounds in the binPMF factors come from. The sentence "**The actual origins of the detected compounds and binPMF factors are not examined in this measurement report.**" is now added to the end of the introduction section to highlight this. Because of that, we also didn't include the map and any more detailed meteorological information of the sampling site. However, we have now included a better description of the sampling site to Sect. 2.1 with the updated text "The measurement site is located in the middle of fields and has the shortest distance to the sea of 500 m **and to the nearest forest of 100 m. The nearest town, Parainen, with ∼15 000 inhabitants, is located 5 km to the west and a larger city, Turku, with ∼195 000 inhabitants, is located ∼18 km toward the inland**." to give an idea on what the sources of the compounds, in addition to the field itself, could be.

**3.** *The time profiles of F7 compounds did not correlate with F8 compounds (Figure 7), I do not understand why the F7 formed from the F8?*

By looking at the time series of F7, F8 and $O_3$ in Fig. 7, it can be observed that when NetRad increases, F8 decreases and almost the same amount is added to F7, while their sum behaves very similar to the trend of the $O_3$ concentration. Therefore, they don't need to correlate for F7 to be formed from F8, but rather to anti-correlate (but only with NetRad).

However, because the actual (chemical) transformation of F8 to F7 (where F8 is the precursor for F7) cannot be proven, we have loosened some wordings related to it. We have now used wordings like "**seemigly** transforms" instead of "transforms" and "acts **like** a precursor" instead of "acts  a precursor" in all corresponding locations to highlight that this transformation is only what has been observed from the time series but the chemical mechanism hasn't been examined. Additionally, the sentence "It is evident that F7(day-N-HOMs) is connected to particle formation and growth process and F8(morning-N-HOMs) acts as a precursor for F7(day-N-HOMs)." in Sect. 4.4 is now updated to "It is evident that F7(day-N-HOMs) is connected to particle formation and growth process and F8(morning-N-HOMs) acts **like** a precursor for F7(day-N-HOMs) **because the F8(morning-N-HOMs) level decreases with increasing F7(day-N-HOMs)**."

See also the referee #1 comment 19, which is related to this as well.

**4.** *Did the authors detect halogenated organics due to the proximity of the measurement site to the sea?*

Halogenated organics were not examined from this nitrate CI-APi-TOF data because the instrument is typically not used to detect them (at least with nitrate ionization) and, according to our knowledge, there is currently no expectation that they would be connected to NPF events.

*Specific comments:*

**5.** *Line 9: "Values of $f_{F7}$ higher than 0.5 were typically observed during the NPF events". However, Figure 7c showed this value is lower than 0.5 during the NPF events on May 8-11 and 17.*

The critical value of $f_{F7}$ for NPF events is not exactly 0.5. Therefore, we have used the "$\sim$0.5" notation. However, the uncertainty of the critical value is now better defined and all "$\sim$0.5" notations are now replaced with "$0.50 \pm 0.05$". Thus, the minimum of the critical value is about 0.45, which is exceeded during the NPF events on the days mentioned in the comment.

**6.** *Line 75: What is the mass resolution of CIMS during the field observation?*

The mass resolution is now mentioned with the added sentence "**The APi-TOF mass spectrometer provided the mass resolving power of 3500–4000 Th/Th for the studied mass range with the used voltage settings.**"

**7.** *Line 108: "Ions smaller than 169 Th were omitted because there are many organic compounds that are unlikely the key compounds in NPF". However, methanesulfonic acid can also efficiently initiate NPF in the presence of small alkylamines and water (Chen et al., 2016; Dawson et al., 2012).*

See the referee #1 comment 8. The citation to the article by Dawson et al. (2012) is now added to the text about methanesulfonic acid and NPF in the introduction section.

*References:*

*Chen, H., Varner, M. E., Gerber, R. B. and Finlayson-Pitts, B. J.: Reactions of Methanesulfonic Acid with Amines and Ammonia as a Source of New Particles in Air, J. Phys. Chem. B, 120(8), 1526–1536, doi:10.1021/acs.jpcb.5b07433, 2016.*

[revised manuscript text omitted]

**Figure S10.** Pearson's correlation coefficient ($R$) between all particle size bins and **(a)** the concentrations of sulfuric acid ([$H_2SO_4$]) and ammonia ([$NH_3$]) and **(b)** binPMF factors 3 and 7, together with the variable $f_{F7}$. Data below 3 nm are from a PSM, over 5 nm are from the DMPS, and the remaining part between them are from the NAIS. Solid lines denote data from the days with NPF events and dashed lines from the days with no NPF event. Due to a relative low amount of the data in these plots, the lines are shown regardless of their statistic significance.

**Table S1.** Classifications of the measurement days in terms of new particle formation events, according to the behavior of the measured particle size distributions.

| Day | Classification | Day | Classification | Day | Classification |
|-----|---------------|-----|---------------|-----|---------------|
| | | May 1 | unclear | Jun 1 | no event |
| | | May 2 | no event | Jun 2 | no event |
| Apr 3 | no event | May 3 | event | Jun 3 | no event |
| Apr 4 | no event | May 4 | unclear | Jun 4 | no event |
| Apr 5 | no event | May 5 | unsure | Jun 5 | no event |
| Apr 6 | no event | May 6 | no event | Jun 6 | no event |
| Apr 7 | no event | May 7 | no event | Jun 7 | no event |
| Apr 8 | no event | May 8 | event | Jun 8 | no event |
| Apr 9 | no event | May 9 | event | Jun 9 | unclear |
| Apr 10 | no event | May 10 | event | Jun 10 | event |
| Apr 11 | event | May 11 | no event | Jun 11 | event |
| Apr 12 | event | May 12 | no event | Jun 12 | unclear |
| Apr 13 | unsure | May 13 | event | Jun 13 | unclear |
| Apr 14 | unsure | May 14 | event | Jun 14 | no event |
| Apr 15 | event | May 15 | event | Jun 15 | no event |
| Apr 16 | event | May 16 | event | Jun 16 | no event |
| Apr 17 | event | May 17 | event | Jun 17 | unclear |
| Apr 18 | event | May 18 | event | Jun 18 | unclear |
| Apr 19 | unclear | May 19 | no event | Jun 19 | unclear |
| Apr 20 | unclear | May 20 | no event | Jun 20 | unclear |
| Apr 21 | no event | May 21 | no event | Jun 21 | unclear |
| Apr 22 | no event | May 22 | unclear | Jun 22 | no event |
| Apr 23 | no event | May 23 | no event | Jun 23 | unclear |
| Apr 24 | no event | May 24 | unclear | Jun 24 | event |
| Apr 25 | unclear | May 25 | no event | Jun 25 | event |
| Apr 26 | no event | May 26 | unclear | | |
| Apr 27 | no event | May 27 | event | | |
| Apr 28 | event | May 28 | no event | | |
| Apr 29 | event | May 29 | unclear | | |
| Apr 30 | event | May 30 | unclear | | |
| | | May 31 | event | | |

**Table S2.** List of the 10 tallest peaks in the binPMF factors. The values denote $m/z$ (in Th) of the peaks in ascending order. The tallest peaks are marked in bold.

| F1 (low-mass) | F2 (water-nitrate dimer) | F3 (nitrophenol, nitrocatechol) | F4 (OOMs) | F5 (mixed) | F6 (water-nitrate monomer) | F7 (day-N-HOMs) | F8 (morning-N-HOMs) |
|---|---|---|---|---|---|---|---|
| 178.0124 | 170.0531 | 182.9971 | 180.0173 | 169.0690 | **170.0531** | **180.0156** | 182.9943 |
| 180.0171 | **179.0170** | **201.0171** | 182.9899 | 170.0254 | 206.0747 | 201.0157 | 201.0159 |
| **182.9960** | 197.0278 | 202.0200 | 208.0300 | **171.0606** | 224.0849 | 208.0159 | 220.0477 |
| 186.9861 | 206.0718 | 217.0109 | **220.0465** | 179.0216 | 242.0948 | 224.0156 | 220.0477 |
| 192.0182 | 215.0387 | 234.0615 | 222.0456 | 179.9922 | 260.1054 | 234.0614 | 239.0192 |
| 194.0262 | 224.0849 | 239.0162 | 234.0535 | 183.0787 | 278.1149 | 248.0509 | 267.0465 |
| 201.0153 | 233.0494 | 241.0585 | 236.0449 | 183.9966 | 296.1236 | 255.0276 | **295.0749** |
| 220.0494 | 251.0581 | 248.0504 | 239.0190 | 185.0550 | 314.1326 | 283.0417 | 323.0654 |
| 239.0191 | 269.0695 | 255.0434 | 248.0564 | 187.0077 | 332.1426 | 323.0650 | 325.0703 |
| 246.0000 | 287.0777 | 264.0093 | 250.0571 | 220.0135 | 350.1475 | 339.0585 | 370.0555 |

**Table S3.** Pearson's correlation coefficients ($R$) between the most important binPMF factors and UMR data. The bold values denote the most suitable UMR tracers for the binPMF factors. A perfectly selective UMR tracer for a factor would have $R$ equal to the ones in footnotes (and $R = 1$ between the factor and the tracer). $R$ for the other factors not shown here for these selected UMR tracers are less than 0.40.

| UMR | 220 | 236 | 285 | 271 | 339 | 295 | 265 |
|---|---|---|---|---|---|---|---|
| F4(OOMs) | **0.98** | **0.97** | $0.65^a$ | $0.64^a$ | $0.59^a$ | $0.43^b$ | $0.62^b$ |
| F7(day-N-HOMs) | $0.54^a$ | $0.63^a$ | **0.91** | **0.89** | **0.87** | $0.37^c$ | $0.54^c$ |
| F8(morning-N-HOMs) | $0.47^b$ | $0.36^b$ | $0.36^c$ | $0.42^c$ | $0.52^c$ | **0.95** | **0.82** |

[a] $R = 0.47$ between F4(OOMs) and F7(day-N-HOMs)

[b] $R = 0.30$ between F4(OOMs) and F8(morning-N-HOMs)

[c] $R = 0.25$ between F7(day-N-HOMs) and F8(morning-N-HOMs)